# PROJECTED LATENT MARKOV CHAIN MONTE CARLO: CONDITIONAL SAMPLING OF NORMALIZING FLOWS

**Chris Cannella, Mohammadreza Soltani & Vahid Tarokh**
Department of Electrical and Computer Engineering
Duke University

## ABSTRACT

We introduce Projected Latent Markov Chain Monte Carlo (PL-MCMC), a technique for sampling from the exact conditional distributions learned by normalizing flows. As a conditional sampling method, PL-MCMC enables Monte Carlo Expectation Maximization (MC-EM) training of normalizing flows from incomplete data. Through experimental tests applying normalizing flows to missing data tasks for a variety of data sets, we demonstrate the efficacy of PL-MCMC for conditional sampling from normalizing flows.

## 1 INTRODUCTION

Conditional sampling from modeled joint probability distributions offers a statistical framework for approaching tasks involving missing and incomplete data. Deep generative models have demonstrated an exceptional capability for approximating the distributions governing complex data. Brief analysis illustrates a fundamental guarantee for generative models: the inaccuracy (i.e. divergence from ground truth) of a generative model's approximated joint distribution upper bounds the expected inaccuracies of the conditional distributions known by the model, as shown in Appendix A. Although this guarantee holds for all generative models, specialized variants are typically used to approach tasks involving the conditional distributions among modeled variables, due to the difficulty in accessing the conditional distributions known by unspecialized generative models. Quite often, otherwise well trained generative models possess a capability for conditional inference that is regrettably locked away from our access.

Normalizing flow architectures like RealNVP (Dinh et al., 2014) and GLOW (Kingma & Dhariwal, 2018) have demonstrated accurate and expressive generative performance, showing great promise for application to missing data tasks. Additionally, by enabling the calculation of exact likelihoods, normalizing flows offer convenient mathematical properties for approaching exact conditional sampling. We are therefore motivated to develop techniques for sampling from the exact conditional distributions known by normalizing flows. In this paper, we propose Projected Latent Markov Chain Monte Carlo (PL-MCMC), a conditional sampling technique that takes advantage of the convenient mathematical structure of normalizing flows by defining a Markov Chain within a flow's latent space and accepting proposed transitions based on the likelihood of the resulting imputation. In principle, PL-MCMC enables exact conditional sampling without requiring specialized architecture, training history, or external inference machinery.

**Our Contributions:** We prove that a Metropolis-Hastings implementation of our proposed PL-MCMC technique is asymptotically guaranteed to sample from the exact conditional distributions known by any normalizing flow satisfying very mild positivity and smoothness requirements. We then describe how to use PL-MCMC to perform Monte Carlo Expectation Maximization (MC-EM) training of normalizing flows from incomplete training data. To illustrate and demonstrate aspects of the technique, we perform a series of experiments utilizing PL-MCMC to complete CIFAR-10 images, CelebA images, and MNIST digits affected by missing data. Finally, we perform a series of experiments training non-specialized normalizing flows to model MNIST digits and continuous UCI datasets from incomplete training data to verify the performance of the proposed method. Through these experimental results, we find that PL-MCMC holds great practical promise for tasks requiring conditional sampling from normalizing flows.

## 2 RELATED WORK

A conditional variant of normalizing flows has been introduced by Lu & Huang (2020) to model a single conditional distribution between architecturally fixed sets of conditioned and conditioning variables. While quite capable of learning individual conditional distributions, conditional variants do not enable arbitrary conditional sampling from a joint model. Richardson et al. (2020) concurrently train a deterministic inference network alongside a normalizing flow for inferring missing data. Although such an inference network can produce deterministic imputations consistent with the distributions learned by a normalizing flow, it cannot stochastically sample from the conditional distributions known by the flow. Li et al. (2019) introduce shared parameter approximations that allow the derivation of approximate conditional normalizing flows, though these approximations do not guarantee exact sampling from the conditional distributions of a particular joint model. Similar techniques for approaching missing data with other generative models, such as generative adversarial networks (GANs) and variational auto-encoders (VAEs), have been introduced with similar limitations (Ivanov et al., 2018; Yoon et al., 2018; Li et al., 2018).

A MCMC procedure for sampling from the conditional distributions of VAEs has been introduced by Rezende et al. (2014) and refined by Mattei & Frellsen (2018). This procedure fundamentally relies on the many-to-many relationship between the latent and modeled data spaces of VAEs, and cannot be directly applied to normalizing flows, wherein the latent state uniquely determines (and is uniquely determined by) the modeled data state. By following an unconstrained Markov Chain within the latent space, PL-MCMC mirrors this VAE conditional sampling procedure within the context of normalizing flows.

PL-MCMC leverages the probabilistic structure learned by a normalizing flow to produce efficient Markov Chains. The utility of the mathematical structure of normalizing flows for approaching Monte Carlo estimation via independence sampling has been demonstrated by Müller et al. (2019). The probabilistic structure of normalizing flows has also been shown to improve unconditional sampling from externally defined distributions by Hoffman et al. (2019). In using this learned structure, we believe that PL-MCMC receives many of the benefits of Adaptive Monte Carlo methods (Haario et al., 2001; Foreman-Mackey et al., 2013; Zhu, 2019), as explained in Appendix B.

PL-MCMC's unconstrained Markov Chain through the latent space is not the only conceivable option for sampling from the conditional distributions described by normalizing flows. As normalizing flows enable exact joint likelihood calculations, we could employ MCMC methods through the modeled data space. Dinh et al. (2014) demonstrate a stochastic conditional MAP inference that can be adapted to implement the unadjusted Langevin algorithm (Fredrickson et al., 2006; Durmus et al., 2019) or the Metropolis adjusted Langevin algorithm (Grenander & Miller, 1994). A constrained Hamiltonian Monte Carlo approach has also been introduced in the context of conditional sampling from generative models by Graham et al. (2017). MCMC methods restricted to the modeled data space approach the normalizing flow as a sort of blackbox oracle to be used only for calculations regarding data likelihood. By design, PL-MCMC leverages the flow's one-to-one mapping between latent and modeled data spaces, thereby taking better advantage of the probabilistic structure learned by our normalizing flows to perform conditional sampling.

## 3 THE PL-MCMC APPROACH

We consider a normalizing flow between latent space $\Xi$ and modeled data space $\mathcal{X}$, defining the mappings $f_\theta : \Xi \mapsto \mathcal{X}$ and $f_\theta^{-1} : \mathcal{X} \mapsto \Xi$. This normalizing flow imposes the probability density $p_{f,\theta}(\mathbf{x})$ onto all modeled data values $\mathbf{x} \in \mathcal{X}$. By the pairing $(\mathbf{x}_M; \mathbf{x}_O)$, we denote the missing and observed portion of a modeled data value with joint density $p_{f,\theta}(\mathbf{x}_M; \mathbf{x}_O)$ under our normalizing flow. Our goal is to sample from the conditional density described by the normalizing flow, $p_{f,\theta}(\mathbf{x}_M|\mathbf{x}_O)$.

### 3.1 THE PROJECTED LATENT TARGET DISTRIBUTION

Rather than targeting the conditional distribution of missing values directly, PL-MCMC targets a distribution of latent variables that, after mapping through the flow's transformation, marginalizes to the desired conditional distribution. Let the Markov Chain be composed of latent state $\boldsymbol{\xi} \in \Xi$,

mapping to the modeled data pair $f_\theta(\boldsymbol{\xi}) = (\mathbf{y}_M; \mathbf{y}_O)$. Let $q$ be an arbitrary smooth density over observed variables, $\mathbf{y}_O$. PL-MCMC targets the distribution whose (unnormalized) density within the modeled data space is $q(\mathbf{y}_O)p_{f,\theta}(\mathbf{y}_M|\mathbf{x}_O)$. Fundamentally, PL-MCMC is a marginal MCMC method (Van Dyk, 2010) that uses the otherwise observed attributes, $\mathbf{y}_O$, as auxiliary working variables to take full advantage of the probabilistic structure learned by the normalizing flow.

## 3.2 DESCRIPTION OF METROPOLIS-HASTINGS PL-MCMC ALGORITHM

For a Metropolis-Hastings implementation of PL-MCMC, we introduce a transition kernel $g(\boldsymbol{\xi}'|\boldsymbol{\xi})$ for generating proposal latent states. We sample a new proposal latent vector $\boldsymbol{\xi}' \sim g(\boldsymbol{\xi}'|\boldsymbol{\xi})$, mapping to the modeled data pair $f_\theta(\boldsymbol{\xi}') = (\mathbf{y}'_M; \mathbf{y}'_O)$. An illustrative diagram of the production of PL-MCMC proposals is provided in Appendix B. This proposal is then accepted with probability:

$$\alpha = \min(1, \ \frac{q(\mathbf{y}'_O)p_{f,\theta}(\mathbf{y}'_M; \mathbf{x}_O)g(\boldsymbol{\xi}|\boldsymbol{\xi}')|\det \frac{\partial f_\theta}{\partial \boldsymbol{\xi}'}|}{q(\mathbf{y}_O)p_{f,\theta}(\mathbf{y}_M; \mathbf{x}_O)g(\boldsymbol{\xi}'|\boldsymbol{\xi})|\det \frac{\partial f_\theta}{\partial \boldsymbol{\xi}}|}).$$

---

**Algorithm 1:** PL-MCMC Metropolis-Hastings Update

**Input:** Observed data $\mathbf{x}_O$, normalizing flow $f_\theta$, modeled joint density $p_{f,\theta}(\mathbf{x}_M; \mathbf{x}_O)$. Latent transition kernel $g(\boldsymbol{\xi}'|\boldsymbol{\xi})$ and auxiliary density $q(\mathbf{y}_O)$. Initial latent state $\boldsymbol{\xi}$

Sample $\boldsymbol{\xi}' \sim g(\boldsymbol{\xi}'|\boldsymbol{\xi})$;
$\mathbf{y}_M; \mathbf{y}_O \leftarrow f_\theta(\boldsymbol{\xi})$;
$\mathbf{y}'_M; \mathbf{y}'_O \leftarrow f_\theta(\boldsymbol{\xi}')$;
$\alpha \leftarrow \min(1, \ \frac{q(\mathbf{y}'_O)p_{f,\theta}(\mathbf{y}'_M; \mathbf{x}_O)g(\boldsymbol{\xi}|\boldsymbol{\xi}')|\det \frac{\partial f_\theta}{\partial \boldsymbol{\xi}'}|}{q(\mathbf{y}_O)p_{f,\theta}(\mathbf{y}_M; \mathbf{x}_O)g(\boldsymbol{\xi}'|\boldsymbol{\xi})|\det \frac{\partial f_\theta}{\partial \boldsymbol{\xi}}|})$;
Sample $u \sim \text{Uniform}[0, 1]$;
**if** $u < \alpha$ **then**
  $\boldsymbol{\xi} \leftarrow \boldsymbol{\xi}'$;

---

## 3.3 THEORETICAL JUSTIFICATION OF THE ALGORITHM

**Proposition.** *For a given* $\mathbf{x}_O$, *if* $g(\boldsymbol{\xi}'|\boldsymbol{\xi})$, $p_{f,\theta}(\mathbf{y}_M; \mathbf{y}_O)$, *and* $q(\mathbf{y}_O)$ *are positive for any choice of* $(\mathbf{y}_M; \mathbf{y}_O) \in \mathcal{X}$ *and* $\boldsymbol{\xi}', \boldsymbol{\xi} \in \Xi$ *and are the densities for absolutely continuous distributions, the PL-MCMC update procedure listed in Algorithm 1 yields a Markov Chain of latent states* $\boldsymbol{\xi}$ *whose corresponding modeled data pairs,* $f_\theta(\boldsymbol{\xi}) = (\mathbf{y}_M; \mathbf{y}_O)$ *, converge to a distribution with* $\mathbf{y}_M$ *having marginal density* $p_{f,\theta}(\mathbf{y}_M|\mathbf{x}_O)$.

*Proof.* Under these assumptions, the diffeomorphism (i.e, an invertible and differentiable mapping) provided by the flow $f_\theta$ allows us to interpret the latent transition kernel $g(\boldsymbol{\xi}'|\boldsymbol{\xi})$ as the transition kernel $g(f_\theta^{-1}(\mathbf{y}')|f_\theta^{-1}(\mathbf{y}))|\det \frac{\partial f_\theta^{-1}}{\partial \mathbf{y}'}|$ within the modeled data space that is positive for all $\mathbf{y}, \mathbf{y}' \in \mathcal{X}$ and is the density for an absolutely continuous distribution. Additionally, we note:

$$\frac{q(\mathbf{y}'_O)p_{f,\theta}(\mathbf{y}'_M; \mathbf{x}_O)}{q(\mathbf{y}_O)p_{f,\theta}(\mathbf{y}_M; \mathbf{x}_O)} = \frac{q(\mathbf{y}'_O)p_{f,\theta}(\mathbf{y}'_M|\mathbf{x}_O)}{q(\mathbf{y}_O)p_{f,\theta}(\mathbf{y}_M|\mathbf{x}_O)}.$$

The diffeomorphism provided by the flow $f_\theta$ also guarantees that $q(\mathbf{y}_O)p_{f,\theta}(\mathbf{y}_M|\mathbf{x}_O)$ is positive for all $(\mathbf{y}_M; \mathbf{y}_O) \in \mathcal{X}$ and is the density for an absolutely continuous distribution. The procedure listed in Algorithm 1 therefore describes a Metropolis-Hastings update satisfying the conditions described by Tsvetkov et al. (2013). The paired values $f_\theta(\xi) = (\mathbf{y}_M; \mathbf{y}_O)$ obtained through iterated application of Algorithm 1 thus converge to a target distribution with (unnormalized) density $q(\mathbf{y}_O)p_{f,\theta}(\mathbf{y}_M|\mathbf{x}_O)$. □

The requirements for convergence are very mild and are satisfied by the most common choices for latent, transition proposal, and auxiliary distributions (e.g. multivariate normal distributions). We note that the eventual convergence of the PL-MCMC update towards the desired conditional

distribution is not influenced by our choice of the auxiliary distribution $q$. However, the choice of this auxiliary distribution can affect the rate of convergence. We have found agreeable performance by selecting $q$ to be an independent normal distribution centered on the conditioning values $\mathbf{x}_O$. This guides the Markov Chain towards reasonable samples more quickly by leveraging learned dependencies between the observed and missing components of the modeled data.

## 4 TRAINING NORMALIZING FLOWS FROM MISSING DATA

With PL-MCMC providing samples from the conditional distributions of normalizing flows, a natural application of the technique is in MC-EM training (Dempster et al., 1977; Wei & Tanner, 1990; Neath et al., 2013) of normalizing flows from incomplete data. MC-EM training involves imputing missing values within the training set via conditional sampling of our current model, and then updating the parameters of our model to best fit the newly imputed training set. As described within Appendix C, this leads to Algorithm 2, with `PL-MCMC`$(\mathbf{x}_{O,i}; p_{f,\theta}, q_i)$ denoting the distribution obtained by following an implementation of PL-MCMC with auxiliary density $q_i$ (defined in 3.1) and `train` being any training procedure that returns flow parameters $\theta$ approximately maximizing the likelihood of a complete data training set. For our experimental tests, `PL-MCMC` is obtained through iterated application of Algorithm 1.

---

**Algorithm 2:** Monte Carlo Expectation Maximization Training of Normalizing Flow

---

**Input:** Incomplete training data $X_{train} = \{\mathbf{x}_{O,1}, \mathbf{x}_{O,2}, \ldots, \mathbf{x}_{O,T}\}$. Auxiliary densities $q_i$.
      Normalizing flow training procedure `train`. Parameterized flow architecture $f_\theta$.
**while** *training* **do**
    **for** $i \leftarrow 1$ **to** $T$ **do**
        |   Sample $\mathbf{y}_{M,i} \sim$ `PL-MCMC`$(\mathbf{x}_{O,i}; p_{f,\theta}, q_i)$;
    **end**
    $X'_{train} = \{(\mathbf{y}_{M,1}; \mathbf{x}_{O,1}), (\mathbf{y}_{M,2}; \mathbf{x}_{O,2}), \ldots, (\mathbf{y}_{M,T}; \mathbf{x}_{O,T})\}$;
    $\theta \leftarrow$ `train`$(f, X'_{train})$;
**end**

---

Intuitively, this procedure relies on conditional inference to "boost" the accuracy of our current model for the joint distribution governing the training data. At each step of Algorithm 2, $X'_{train}$ represents samples from an approximation of the modeled data's ground truth distribution. We fit $\theta$ to model this approximate joint distribution. After conditional inference with the new normalizing flow using PL-MCMC, the next iteration of $X'_{train}$ represents samples from a distribution with a smaller divergence from the ground truth distribution, as discussed in Appendix A. Importantly, this MC-EM training procedure assumes that data is missing at random (Little & Rubin, 2019).

## 5 QUALITATIVE EXPERIMENTAL RESULTS

For a qualitative examination of the performance of PL-MCMC, we focus on conditionally sampling missing data using normalizing flows that have been trained from complete data. We must note that the the purpose of PL-MCMC is to sample from a model's conditional distributions, which may not coincide with accurately replicating the ground truth values of missing data. These qualitative experiments are therefore intended to illustrate aspects of the operation of PL-MCMC and to provide a visual verification of the method's performance. Further details of these experiments and examples of unconditioned samples from the normalizing flows are provided in Appendix D.

### 5.1 CONDITIONAL INFERENCE WITH CIFAR-10 IMAGES

We first consider sampling a missing central quarter of CIFAR-10 (Krizhevsky et al., 2009) images ($32 \times 32$ full color images) using a normalizing flow following the GLOW architecture (Kingma & Dhariwal, 2018). To bolster our claim that PL-MCMC does not require specially trained models, we utilize a publicly available pre-trained model (van Amersfoort, 2019) for this experiment. Initial and final completions provided by the Markov Chain are illustrated in Figure 1.

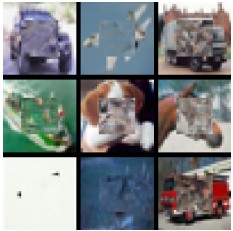 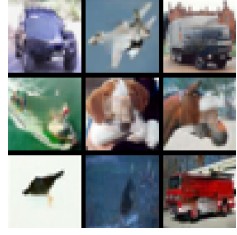 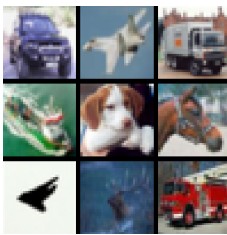

(a) Initial Completions      (b) Final Completions      (c) Ground Truth

Figure 1: Conditional inference of CIFAR-10 images with normalizing flow trained on complete data.

The initial state of the Markov Chain is constructed by filling pixels with RGB values randomly selected from the observed subset. Latent space transitions are generated via small perturbations within the absolute coordinates of the latent space. PL-MCMC is carried out for 25,000 proposals. Example progressions of completions are provided in Figure 2. In comparison with unconditioned samples, the PL-MCMC completions appear reasonable, given the capabilities of the underlying model, and highlight the perceptual benefit provided by conditioned sampling.

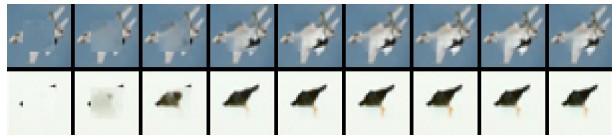

Figure 2: Progression of CIFAR-10 completions over intervals of 3,000 PL-MCMC proposals.

## 5.2 CONDITIONAL INFERENCE WITH CELEBA IMAGES

Next we consider sampling a missing right half of CelebA (Liu et al., 2015) images (aligned, cropped, and resized to $64 \times 64$ full color images) using a normalizing flow following the GLOW architecture (Kingma & Dhariwal, 2018). To bolster our claim that PL-MCMC does not require specially trained models, we utilize a publicly available pre-trained model (Yuki-Chai, 2019) for this experiment. Initial and final completions provided by the Markov Chain are illustrated in Figure 3.

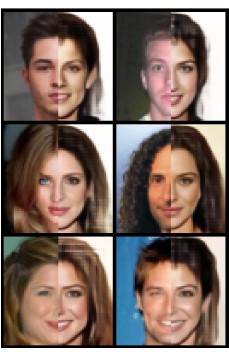 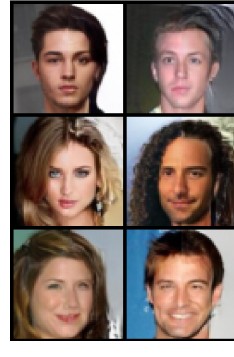 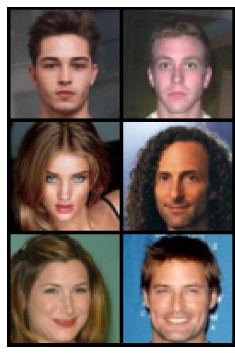

(a) Initial Completions      (b) Final Completions      (c) Ground Truth

Figure 3: Conditional inference of CelebA images with normalizing flow trained on complete data.

The initial state of the Markov Chain is constructed by sampling from the normalizing flow at reduced variance. Latent space transitions are generated via small perturbations within relative coordinates of the latent space. PL-MCMC is carried out for 25,000 proposals. Example progressions of completions are provided in Figure 4. The progression of PL-MCMC completions clearly demonstrates how defining a Markov Chain through the flow's latent space encourages proposing alterations to semantically meaningful attributes.

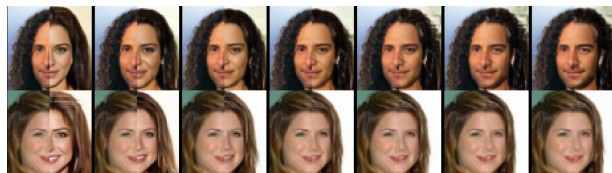

Figure 4: Progression of CelebA completions over intervals of 1000 PL-MCMC proposals.

## 5.3 CONDITIONAL INFERENCE WITH MNIST DIGITS

Finally, we consider sampling missing portions of MNIST (LeCun et al., 1998) digits ($28 \times 28$ monochrome images) using a normalizing flow following a variant of the NICE architecture (Dinh et al., 2014) under a variety of data missingness mechanisms. The missingness mechanisms considered are independent missingness (I.M.), patch missingness (P.M.), and square observation (S.O.), at missingness rates of 0.6, 0.6, and 0.8 respectively. Final completions and conditional expectations as obtained by averaging the final completions of 20 independent PL-MCMC chains are illustrated in Figure 5.

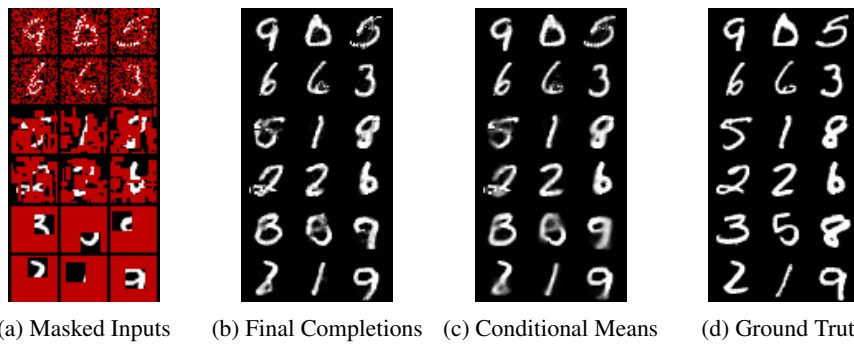

(a) Masked Inputs    (b) Final Completions    (c) Conditional Means    (d) Ground Truth

Figure 5: Conditional inference of MNIST digits with normalizing flow trained on complete data.

The initial state of the Markov Chain is constructed by sampling from the normalizing flow at reduced variance. Latent space transitions are generated by a mixture of small perturbations within the absolute coordinates of the latent space and resampling at reduced variance. PL-MCMC is performed over 2,000 proposals. Example progressions of completions are provided in Figure 6.

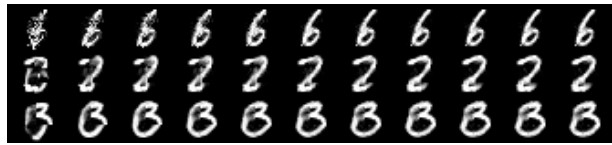

Figure 6: Progression of MNIST completions over intervals of 200 PL-MCMC proposals.

## 6 QUANTITATIVE EXPERIMENTAL RESULTS

As an analytical description of the conditional distributions of non-specialized normalizing flows is infeasible, it is difficult to quantify how well PL-MCMC succeeds in sampling from its intended distributions. Given the extreme dependence of Algorithm 2 on accurate conditional sampling from PL-MCMC for effective training, we therefore quantify the performance of normalizing flows trained from incomplete data as an indication for whether PL-MCMC produces sufficiently accurate and efficient samples to remain useful for real-world missing data tasks. We also test the sampling efficiency of PL-MCMC independently of considerations regarding sampling accuracy. Further details of these experiments are provided in Appendix E.

## 6.1 TRAINING FROM INCOMPLETE MNIST DIGITS

In this experiment, we consider training models of MNIST digits from training sets affected by a variety of data missingness mechanisms and imputing test sets affected by the same missingness mechanisms. The data missingness mechanisms used are are independent missingness (I.M.), patch missingness (P.M.), and square observation (S.O.), with missingess rates of $0.3, 0.6$, and $0.9$. As imputation performance measures, we consider per-pixel reconstruction RMSE and Fréchet Inception Distance (Heusel et al., 2017). As comparison, we include results for imputing using pixel wise observed means and using the convolutional variant of MisGAN (Li et al., 2018). Our normalizing flow is a variant of the NICE architecture. We performed MC-EM training of the normalizing flow for a total of 1,000 epochs. Inference with normalizing flows is performed using a PL-MCMC chain of $2,000$ proposals. Our reported results within Table 1 reflect performance across fifteen distinct pairings of training and test sets (models trained, where applicable, from three distinct training sets and each tested on five distinct test sets). For PL-MCMC, our results reflect imputation performance using individual conditional samples (Ind.) and using the average of 10 conditional samples (Avg.) for test set completion.

Table 1: Comparison of imputation performance for reconstructing MNIST digits. Value means are reported to at most the first significant digit of standard error.

| | Rate | Reconstruction RMSE | | | | FID | | | |
| | | Mean | PL-MCMC Ind. | PL-MCMC Avg. | MisGAN | Mean | PL-MCMC Ind. | PL-MCMC Avg. | MisGAN |
|---|---|---|---|---|---|---|---|---|---|
| I.M. | 0.3 | 0.2570(1) | 0.153(1) | 0.130(2) | 0.1277(4) | 23.5(1) | 1.56(7) | 1.58(8) | 0.17(1) |
| | 0.6 | 0.2573(1) | 0.1585(6) | 0.1456(1) | 0.167(2) | 72.2(1) | 5.7(5) | 6.1(5) | 0.78(2) |
| | 0.9 | 0.2574(0) | 0.261(2) | 0.256(1) | 0.326(4) | 114.7(1) | 87(2) | 90(2) | 11(1) |
| S.O. | 0.3 | 0.0577(3) | 0.0410(3) | 0.0371(3) | 0.0439(6) | 0.1(0) | 0.075(1) | 0.076(1) | 0.006(1) |
| | 0.6 | 0.1688(2) | 0.152(3) | 0.137(2) | 0.159(2) | 5.8(1) | 1.4(2) | 1.7(2) | 0.6(1) |
| | 0.9 | 0.2467(1) | 0.2595(8) | 0.2535(7) | 0.322(1) | 68.7(2) | 50(2) | 54(1) | 4(1) |
| P.M. | 0.3 | 0.2629(3) | 0.1795(8) | 0.1565(5) | 0.1956(8) | 17.0(1) | 1.6(1) | 1.8(1) | 0.8(1) |
| | 0.6 | 0.2641(1) | 0.221(4) | 0.205(3) | 0.247(1) | 57.6(1) | 15(1) | 16(1) | 2.9(2) |
| | 0.9 | 0.2622(0) | 0.2675(8) | 0.2648(9) | 0.3693(9) | 110.5(1) | 89(2) | 92(2) | 16(2) |

As RMSE and FID score are measures of distortion and divergence, respectively, a single imputation estimate cannot simultaneously optimize both (Blau & Michaeli, 2018). MisGAN primarily focuses on minimizing imputation FID, while our MC-EM training favors reducing reconstruction RMSE. Our results highlight a potential advantage of performing imputation via sampling from conditional distributions. With its deterministic imputation procedure, MisGAN is dedicated to minimizing FID and cannot reduce reconstruction RMSE by averaging multiple reconstructions. With PL-MCMC sampling, we can choose, to some degree, whether to minimize FID by imputing with a single sample from the flow's conditional distribution or to minimize RMSE by averaging across multiple samples. These results demonstrate that PL-MCMC is able to sample from the conditional distributions of normalizing flows sufficiently well to acceptably train normalizing flows from MNIST digits affected by a variety of data missingness mechanisms and rates.

## 6.2 TRAINING FROM INCOMPLETE UCI DATASETS

In this experiment, we consider training models of various continuous UCI datasets (Bache & Lichman, 2013) affected by 50% uniformly missing values. As a performance measure, we consider normalized MSE of imputing missing values within the training set. As comparison, we include results for imputing using variable-wise observed means, using the missForest (Stekhoven & Bühlmann, 2012) R package with default settings, and using VAEs via MIWAE (Mattei & Frellsen, 2019). Our normalizing flow is a variant of the NICE architecture. We performed MC-EM training of the normalizing flow for a total of 1,000 epochs. For inference, the PL-MCMC chain is run for 1,000 proposals. Our reported results within Table 2 reflect performance across five distinct training sets. For PL-MCMC, our results reflect imputation performance using individual conditional samples (Ind.) and using the average of 25 conditional samples (Avg.) for test set completion.

Table 2: Comparison of imputation NMSE results for continuous UCI datasets affected by 50% uniform missingness. Value means are reported to at most the first significant digit of standard error.

|  | banknote | breast | concrete | red-wine | white-wine | yeast |
|---|---|---|---|---|---|---|
| PL-MCMC Ind. | 1.12(5) | 0.46(2) | 1.22(4) | 1.22(3) | 1.45(3) | 1.67(5) |
| PL-MCMC Avg. | 0.58(3) | 0.31(2) | 0.67(3) | 0.69(3) | 0.76(1) | 0.96(6) |
| MIWAE | 0.56(4) | 0.29(1) | 0.63(3) | 0.66(2) | 0.73(3) | 0.95(5) |
| missForest | 0.74(3) | 0.31(1) | 0.67(2) | 0.74(3) | 0.81(1) | 1.18(3) |
| Mean | 0.99(1) | 1.00(3) | 1.00(1) | 1.00(2) | 1.01(1) | 0.96(6) |

In all cases, the MC-EM trained normalizing flows perform at least as well as missForest and closely match MIWAE for estimating conditional expectations. We can conclude that, while there is some potential room for improvement in capturing the exact ground truth conditional distributions, MC-EM training of normalizing flows with PL-MCMC produces imputations comparable to those from current methods for this particular task.

## 6.3 SAMPLING EFFICIENCY FOR INFERENCE OF MNIST DIGIT

Here we consider the task of estimating the conditional expectation for the missing region of a single MNIST digit using the average of 100 independent Markov Chains. We also use this experiment as an opportunity to explore the effect on conditional sampling performance produced by different choices for PL-MCMC's auxiliary distribution and the transition proposal distribution. The RMSE versus proposal number of conditional means estimated via Gibbs sampling within the modeled data space and PL-MCMC with varying auxiliary distributions are compared in Figure 7. Statistics are gathered from 10 distinct replications of the experiment.

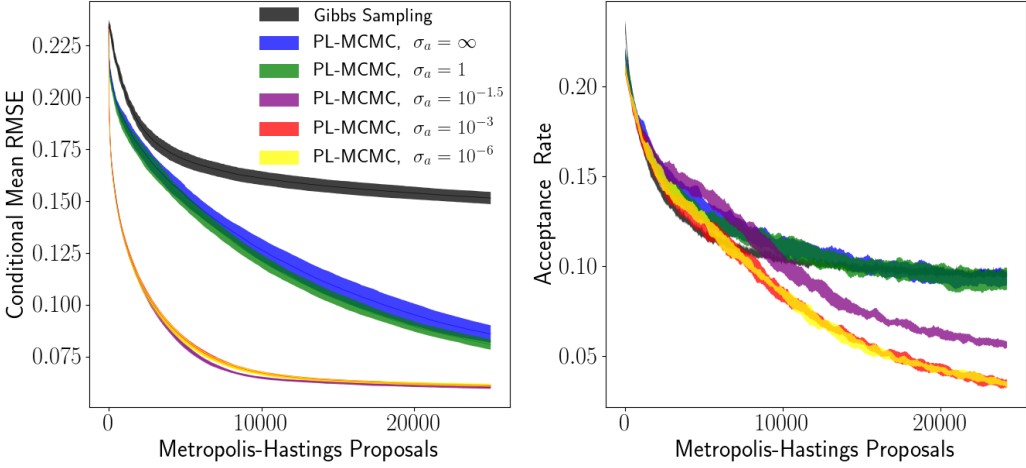

Figure 7: Single standard deviation envelopes of estimated conditional mean per-pixel RMSE and proposal acceptance rate for conditional sampling of MNIST digit. PL-MCMC implementations only differ by choice of auxiliary density.

These results demonstrate that PL-MCMC can offer significant performance gains over comparable MCMC methods confined to the modeled data space. Even when using an improper uniform distribution as the auxiliary density (effectively omitting $q$ from the acceptance probability calculation in Algorithm 1), PL-MCMC can accelerate conditional sampling by leveraging the flow's latent space to propose more effective proposal transitions. Depending on the characteristics of the normalizing flow's conditional distribution, selecting a more restrictive auxiliary distribution can greatly accelerate sampling even further. As the results with auxiliary distributions with standard deviations of $\sigma_a = 10^{-3}$ and $\sigma_a = 10^{-6}$ closely overlap, there may be some concern that the auxiliary distri-

bution might dominate PL-MCMC's behavior and reduce the procedure to a simple search in the latent space to best rebuild the observed data, starting around $\sigma_a = 10^{-3}$. While this concern may be warranted when using exceedingly strong choices for the auxiliary distribution, analysis demonstrates (Appendix E.4) that this is not the case for our results with $\sigma_a = 10^{-3}$. The RMSE versus proposal number of conditional means estimated via Gibbs sampling within the modeled data space and PL-MCMC with varying transition proposal distributions are compared in Figure 8.

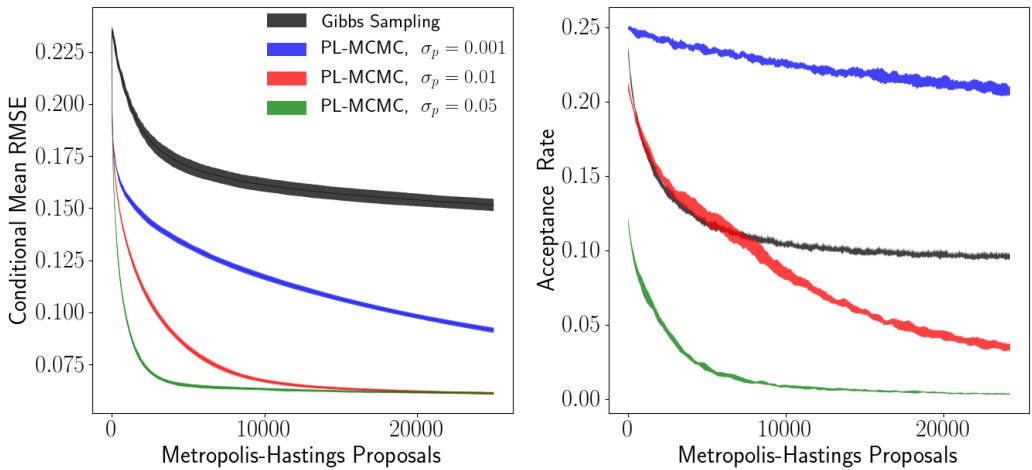

Figure 8: Single standard deviation envelopes of estimated conditional mean per-pixel RMSE and proposal acceptance rate for conditional sampling of MNIST digit. PL-MCMC implementations only differ by scale of perturbations used in their transition proposals.

From these experiments, we offer a few preliminary conclusions regarding the effect of the auxiliary and transition proposal distributions on conditional sampling performance with PL-MCMC. In the Metropolis-Hastings implementation of PL-MCMC, the transition proposal distribution behaves much as one would expect for the transition proposal distributions of any Metropolis-Hastings procedures. Increasing the proposal distribution's scale will accelerate initial convergence, but may encounter problems traversing concentrated regions of the target distributions. To some target distribution dependent point (around $\sigma_a = 10^{-1.5}$ in Figure 7), strengthening the auxiliary distribution will continue to accelerate initial sampling. Beyond this point, further strengthening of the auxiliary distribution can be detrimental to sampling performance, as the auxiliary distribution becomes mismatched to the intrinsic coupling between missing and observed values modeled by the flow's conditional distribution. Additional comparisons, including comparisons with respect to approximate computational cost, are provided within Appendix E.

## 7 CONCLUSION AND FUTURE WORK

The mathematical structure of normalizing flows is exceptionally convenient for approaching conditional sampling via MCMC. By leveraging this mathematical structure, our proposed PL-MCMC technique enables asymptotically exact conditional inference with normalizing flows, without requiring specialized architecture, training history, or external inference machinery. The particular implementations used in our experiments are primarily intended to serve as proof-of-concept illustrations of the PL-MCMC technique. Further research would be necessary to determine optimal choices of auxiliary distributions,transition proposal distributions, and MC-EM training procedures. Sampling performance may be improved by replacing Metropolis-Hastings proposals with a more sophisticated technique, such as Hamiltonian Monte Carlo. Our experimental results demonstrate that, even when implemented with a naive Metropolis-Hastings procedure, PL-MCMC enables effective sampling from its intended distributions under practical settings. We believe that, with the PL-MCMC technique, normalizing flows hold great promise for approaching missing data tasks.

ACKNOWLEDGEMENTS

This work was supported by the Office of Naval Research Grant No. N00014-18-1-2244.

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

## A    KNOWING THE JOINT DISTRIBUTION IMPLIES KNOWING THE CONDITIONAL DISTRIBUTIONS

In this section, we establish a fundamental guarantee regarding the accuracy of the conditional distributions known by generative models. Consider two random variables, $x$ and $y$, with ground truth joint distribution $p_{x,y}$ that we have approximated by a generative model with joint distribution $q_{x,y}$. In the discrete case, we may expand through the definition of the Kullback-Liebler divergence between our generative model and the ground truth to find that:

$$
\begin{aligned}
D_{KL}(p_{x,y}||q_{x,y}) &= -\sum_{x,y} p(x,y) \log\ q(x,y) + \sum_{x,y} p(x,y) \log\ p(x,y) \\
&= \mathbb{E}_x[-\sum_y p(y|x) \log\ q(y|x)q(x) + \sum_y p(y|x) \log\ p(y|x)p(x)] \\
&= \mathbb{E}_x[D_{KL}(p_{y|x}||q_{y|x}) - \log\ q(x) + \log\ p(x)] \\
&= \mathbb{E}_x[D_{KL}(p_{y|x}||q_{y|x})] + D_{KL}(p_x||q_x).
\end{aligned}
$$

From the non-negativity of the Kullback-Liebler divergence, we are then guaranteed:

$$
D_{KL}(p_{x,y}||q_{x,y}) \geq \mathbb{E}_x[D_{KL}(p_{y|x}||q_{y|x})].
$$

With equality only when our generative model perfectly models the marginal distribution of $x$. When considering the task of inferring $y$ from observed values of $x$, the expected performance of our generative model in approximating the conditional distribution of $y$ given $x$ is no worse than its performance in approximating the full joint distribution between $x$ and $y$. Therefore, if we know that a generative model is a good approximation of the joint distribution governing some set of random variables, then it must also know good approximations of the conditional distributions among those random variables.

This inequality also serves as a justification for Monte Carlo Expectation Maximization training. When using our modeled distribution $q_{x,y}$ to impute an incomplete training set, the newly imputed training set is sampled from the distribution $q_{y|x}p_x$. We can easily see that:

$$
D_{KL}(p_{x,y}||q_{y|x}p_x) = \mathbb{E}_x[D_{KL}(p_{y|x}||q_{y|x})] \leq D_{KL}(p_{x,y}||q_{x,y}).
$$

In the asymptotic limit of dataset size, conditionally inferring missing values within the dataset results in samples from a distribution whose divergence from ground truth is no worse than that of the original model. Assuming that the original model describes the distribution of a previously imputed version of the training set, this implies that our newly training set is at least as reflective of the ground truth distribution as the previous training set. In practice, we find that conditional imputation tends to improve divergence of the training set, which in turn allows MC-EM training to improve our model of the joint distribution.

## B    THE ADVANTAGE OF LATENT SPACE PROPOSALS

Here, we relay our intuition regarding the advantages of defining a Markov Chain within the latent space of a normalizing flow. This section provides a heuristic argument and therefore utilizes informal terminology to convey our current understanding. Take a normalizing flow between latent space $\Xi$ and modeled data space $\mathcal{X}$, defining the mappings $f_\theta : \Xi \mapsto \mathcal{X}$ and $f_\theta^{-1} : \mathcal{X} \mapsto \Xi$. This normalizing flow imposes the probability density $p_{f,\theta}(\mathbf{x})$ onto all modeled data values $x \in \mathcal{X}$. As practical applications of normalizing flows primarily involve data embedded within a euclidean space, we will confine this discussion to scenarios where latent and modeled data values are both points in $\mathbb{R}^n$ for some $n$. In these cases, it is straightforward to discuss neighborhoods of fixed radius around points within both the latent and modeled data spaces.

For now, let us consider the task of forming a Markov Chain for unconditionally sampling from the density $p_{f,\theta}(\mathbf{x})$. For simplicity, let us only consider proposal perturbations within some fixed

radius of the Markov Chain's current state. With the one-to-one mapping provided by the flow, we have the option of tracking and perturbing the current state within either the latent space or the modeled data space. When considering the neighborhood of data points, probability mass within the modeled data space is often non-isotropic for highly structured data. However, probability mass is nearly isotropic within the latent space in the neighborhood of the latent representations of data points, assuming that the distribution on latent space states has been appropriately chosen (as is the case for the commonly used multivariate normal or logistic distributions). As a result, performing an isotropic perturbation within the latent space results in proposals that are about as likely as the starting state. Within the modeled data space, even a very small isotropic perturbation can produce proposals that are far more unlikely than the starting state. As an example, suppose our normalizing flow was well trained to model a set of high-fidelity images. If our proposals within the modeled data space were created by adding independent Gaussian perturbations to pixel values, we would almost always inject noise into the image and proposals within the modeled data space would be tend to be unlikely, low-fidelity images. With the assumption that transitions between equally likely states are usually accepted and transitions to much more unlikely states are usually rejected, we should expect latent space proposals to be accepted more frequently compared to modeled data space proposals.

As an intuition, we could say that perturbations within the flow's latent space are semantically meaningful for the modeled data set. As demonstrated by Hoffman et al. (2019), the normalizing flow inherently transforms the modeled probability distribution in a manner that is well suited to exploration using naive, isotropic proposals. This is related to Adaptive Monte Carlo methods (Haario et al., 2001; Foreman-Mackey et al., 2013; Zhu, 2019), which attempt to transform the proposal density to most effectively explore a fixed distribution. With latent space transitions in normalizing flows, it is as though the modeled data distribution has been transformed so as to be best explored by a fixed proposal density.

With PL-MCMC we are concerned with making effective proposals with respect to a conditional distribution. Even when attempting to sample a conditional distribution, utilizing latent space proposals remains beneficial. Define a projection operator via $proj_{\mathbf{x}_O}(\mathbf{y}_M; \mathbf{y}_O) = \mathbf{y}_M; \mathbf{x}_O$, which simply replaces the observed component of a $\mathbf{y} \in \mathcal{X}$ with the conditioning values $\mathbf{x}_O$. The elements of a proposed transition within a Metropolis-Hastings implementation of PL-MCMC are illustrated in Figure 9.

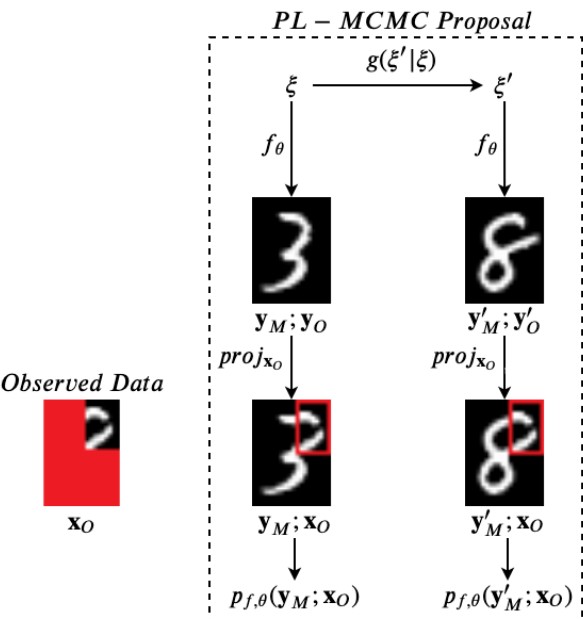

Figure 9: A Metropolis-Hastings PL-MCMC proposal when inferring from observed data $\mathbf{x}_O$.

Averaging over possible pieces of observed data $\mathbf{x}_O$, we expect to find that the set of likely completions, $\mathbf{x}_M$, under the conditional density $p_{f,\theta}(\mathbf{x}_M|\mathbf{x}_O)$ remains essentially a subset of the set

of likely values under the marginalized density $p_{f,\theta}(\mathbf{x}_M)$. If a proposed $\mathbf{x}_M$ is unlikely under $p_{f,\theta}(\mathbf{x}_M)$, we expect it to be unlikely under $p_{f,\theta}(\mathbf{x}_M|\mathbf{x}_O)$. Hence, with PL-MCMC, the advantages of latent space proposals carry through to the conditional inference setting, as the resulting proposed completions can remain likely under $p_{f,\theta}(\mathbf{x}_M)$.

Returning to the example of modeling a set of high-fidelity images, suppose we observe the left half of these images. In general, we would believe that the set of likely right half completions conditioned on the observed left half is well covered by set of likely right halves that we see across the entire distribution of images. Perturbing the pixel values of the right half injects noise, tending to produce a low-fidelity right half, which results in an unlikely, noisy image when combined with the observed left half. Following the PL-MCMC latent perturbations, proposed right halves may be more able to at least remain the high-fidelity right halves of high-fidelity images.

By employing latent space proposals to sample from $p_{f,\theta}(\mathbf{x}_M|\mathbf{x}_O)$, PL-MCMC can more easily propose completions $\mathbf{x}_M$ that could plausibly have been taken from likely members of the modeled data distribution. Of course, for conditional inference, we also need to produce samples that are well matched to the observed data. While latent space proposals assist in making meaningful and efficient transitions within a Markov Chain, PL-MCMC ultimately relies on the auxiliary distribution, $q$, and guaranteed convergence to the correct conditional distribution to effectively sample from typical completions of the observed data.

## C  DETAILS OF MC-EM TRAINING

In this section we review the derivation of Monte Carlo Expectation maximization (Dempster et al., 1977; Wei & Tanner, 1990; Neath et al., 2013) in the context of its use with PL-MCMC. Suppose we are presented with a training set of $T$ observed values (not all missing the same entries), $X_{train} = \{\mathbf{x}_{O,1}, \mathbf{x}_{O,2}, \ldots, \mathbf{x}_{O,T}\}$. Ideally, under the assumption that data values are missing at random (Little & Rubin, 2019), we'd wish to find the flow parameters $\theta$ that maximize the log-likelihood of $X_{train}$:

$$\log p_{f,\theta}(X_{train}) = \sum_{i=1}^{T} \log \Big( \int_{\mathbf{x}_{M,i}} p_{f,\theta}(\mathbf{x}_{M,i}; \mathbf{x}_{O,i}) d\mathbf{x}_{M,i} \Big).$$

Yet the complexity of the normalizing flow makes an analytical computation of the marginal likelihoods of observed data entirely impractical. We therefore utilize the Expecation-Maximization (EM) algorithm (Dempster et al., 1977) to approach this optimization. Following Dempster et al. (1977), we define $Q(\theta'|\theta)$ to be:

$$Q(\theta'|\theta) = \sum_{i=1}^{T} \mathbb{E}_{p_{f,\theta}}[\log( p_{f,\theta'}(\mathbf{x}_{M,i}; \mathbf{x}_{O,i}) )| \mathbf{x}_{O,i}].$$

PL-MCMC can be immediately applied to approximate these expectations. Let the set $Y = \{\mathbf{y}_{M,1}, \mathbf{y}_{M,2}, \ldots, \mathbf{y}_{M,T}\}$ be created by sampling each $\mathbf{y}_{M,i} \sim p_{f,\theta}(\mathbf{y}_{M,i}|\mathbf{x}_{O,i})$ using a PL-MCMC chain as described previously. We may now use the approximation:

$$Q(\theta'|\theta) \approx \sum_{i=1}^{T} \log( p_{f,\theta'}(\mathbf{y}_{M,i}; \mathbf{x}_{O,i}) ).$$

In principle, we would then update $\theta$ following;

$$\theta \leftarrow \operatorname*{argmax}_{\theta'} Q(\theta'|\theta).$$

In practice, it is more feasible to continue to train the flow on the conditionally imputed version of $X_{train}$. With $X'_{train} = \{(\mathbf{y}_{M,1}; \mathbf{x}_{O,1}), (\mathbf{y}_{M,2}; \mathbf{x}_{O,2}), \ldots, (\mathbf{y}_{M,T}; \mathbf{x}_{O,T})\}$ denoting our newly imputed training set and $\texttt{train}$ being any training procedure that returns flow parameters $\theta$ approximately maximizing the likelihood of a complete data training set, we rely on the approximation that:

$$\underset{\theta'}{\arg\max}\, Q(\theta'|\theta) \approx \texttt{train}(f,\, X'_{train}).$$

This approximation immediately leads to our described algorithm for the MC-EM training of normalizing flows using PL-MCMC.

## D DETAILS REGARDING QUALITATIVE EXPERIMENTS

### D.1 DETAILS REGARDING CONDITIONAL INFERENCE WITH CIFAR-10 IMAGES

In this experiment, we infer a missing central quarter (an $8 \times 8$ pixel square) of CIFAR-10 (Krizhevsky et al., 2009) images. The CIFAR-10 dataset is composed of $60,000$ full color $32 \times 32$ images of 10 distinct classes of objects, with $6,000$ images provided for each class. The standard training and test set split for the CIFAR-10 dataset is $50,000$ and $10,000$ images, respectively.

Our chosen normalizing flow is a variant of the GLOW (Kingma & Dhariwal, 2018) architecture. We utilized a publicly available, pre-trained model (van Amersfoort, 2019) for this experiment. In the terminology of Kingma & Dhariwal (2018), the model has a depth of flow (K) of 32 and a total of 3 levels (L) and flow layers utilize $512$ hidden channels. The model was reportedly trained for a total of $1,500$ epochs using Adamax with a learning rate of $5 \times 10^{-4}$ and a batchsize of $64$. We presume, but cannot guarantee, that the model was trained on the standard $50,000$ example CIFAR-10 training set. Examples of unconditioned samples from this model are provided within Figure 10, as obtained with the standard sampling variance, $\sigma = 1.0$ (temperature $T = 1.0$, in the terminology of Kingma & Dhariwal (2018)). From these unconditioned samples, it is clear that the model has not collapsed to memorizing the training set.

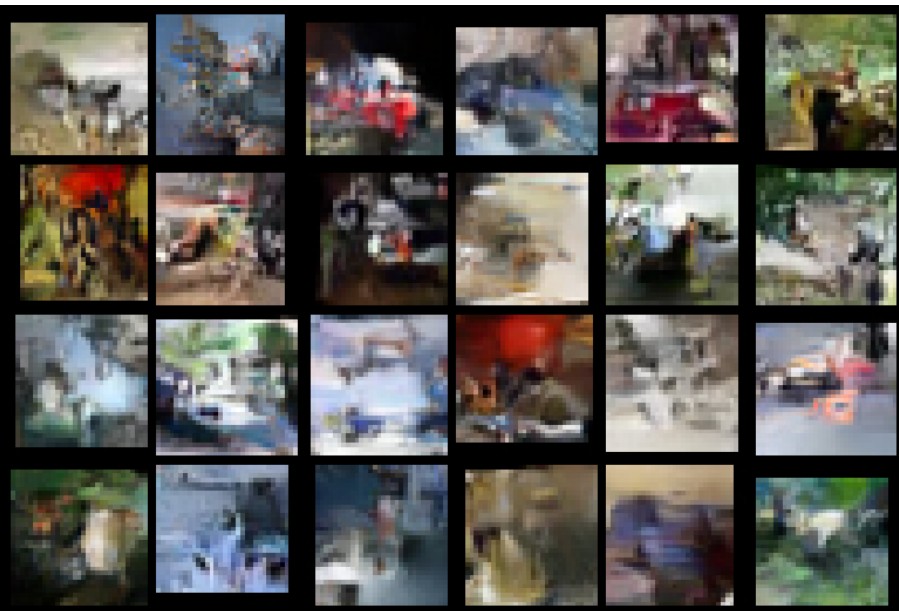

Figure 10: Unconditioned samples at standard variance ($\sigma = 1.0$) from CIFAR-10 model.

The particular implementation of the normalizing flow most easily provided access to a coordinate dependent representation of the latent space, which we call absolute coordinates for the latent space. Fundamentally, the Markov Chain within PL-MCMC may utilize any convenient representation of the latent space, so long as a diffeomorphism still maps that representation back to the modeled data. For this CIFAR-10 experiment, we chose to employ a Markov Chain within the architecture's absolute coordinates. As a general note, the qualifier "absolute" merely refers to the representation favored by the flow's implementation, while the qualifier "relative" refers to the representation best coinciding with the chosen prior distribution for the flow. The terms only reflect aspects of our prac-

tical usage of the representations, as there is no theoretically favored diffeomorphic representation of the latent space.

For inference, we selected images from the standard CIFAR-10 test set. During inference with PL-MCMC, latent space transitions are generated within the above mentioned absolute coordinates of the flow's latent space by small perturbations from the current latent state. When perturbing the latent state, proposals are generated following a perturbation kernel, $g_p(\boldsymbol{\xi}'|\boldsymbol{\xi})$, such that $\boldsymbol{\xi}' \sim \mathcal{N}(\boldsymbol{\xi}, \sigma_p^2 I)$. The auxiliary distribution, $q$, is chosen to target $\mathbf{y}_O \sim \mathcal{N}(\mathbf{x}_O, \sigma_a^2 I)$. For this experiment, we select $\sigma_p = 0.01$ and $\sigma_a = 1 \times 10^{-3}$. PL-MCMC is carried out over 25,000 proposals.

## D.2 Details Regarding Conditional Inference with CelebA Images

In this experiment, we infer a missing right half (a $64 \times 32$ pixel rectangle) of CelebA (Liu et al., 2015) images. The CelebA dataset is composed of $202,599$ full color images of celebrity faces. We utilize the aligned and cropped version of the CelebA dataset, resized to a size of $64 \times 64$.

Our chosen normalizing flow is a variant of the GLOW (Kingma & Dhariwal, 2018) architecture. We utilized a publicly available, pre-trained model (Yuki-Chai, 2019) for this experiment. In the terminology of Kingma & Dhariwal (2018), the model has a depth of flow (K) of 32 and a total of 3 levels (L) and flow layers utilize 512 hidden channels. The model was reportedly trained for a total of $1,500$ epochs using Adamax with a learning rate of $1 \times 10^{-3}$ and a batchsize of 12. We believe that this model was trained on the entirety of the CelebA dataset, with no withheld test or validation set. Examples of unconditioned samples from this model are provided within Figures 11 and 12, as obtained with reduced and standard sampling variance, $\sigma = 0.5$ and $\sigma = 1.0$ respectively (temperatures $T = 0.5$ and $T = 1.0$, in the terminology of Kingma & Dhariwal (2018)). From these unconditioned samples, it is clear that the model has not collapsed to memorizing the training set.

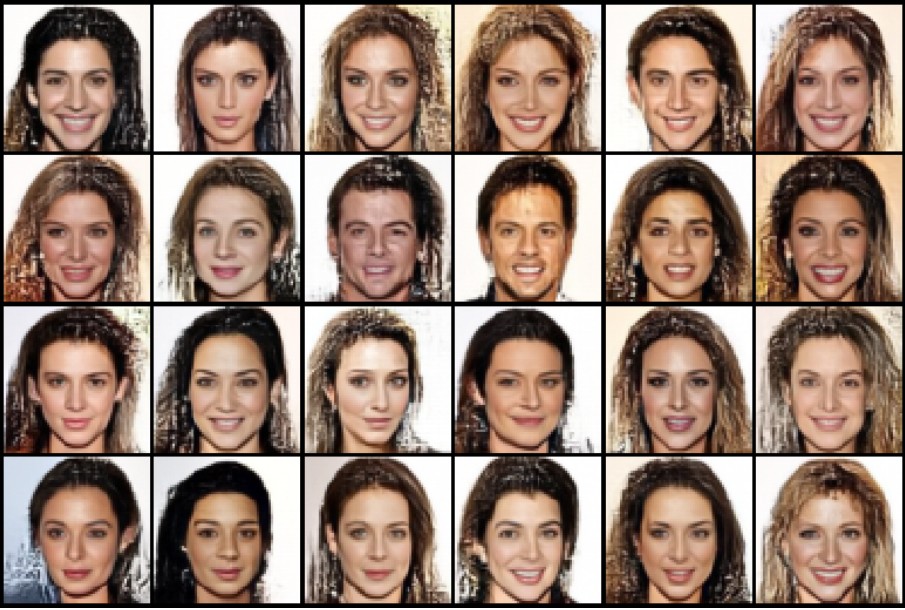

Figure 11: Unconditioned samples at reduced variance ($\sigma = 0.5$) from CelebA model.

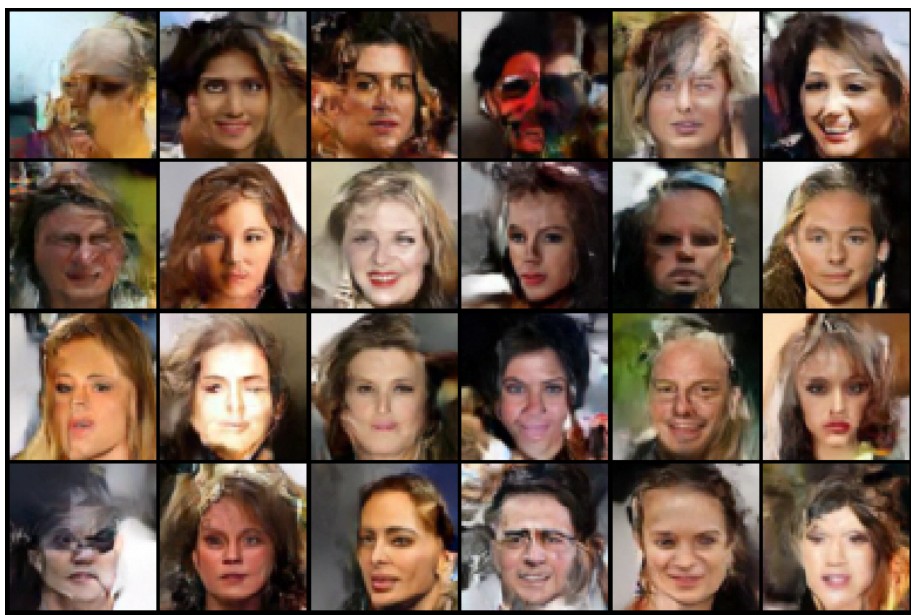

Figure 12: Unconditioned samples at standard variance ($\sigma = 1.0$) from CelebA model.

As in the CIFAR-10 experiment, the particular implementation of the normalizing flow most easily provided access to a coordinate dependent representation of the latent space, which we call absolute coordinates for the latent space. For this CelebA experiment, we chose to employ a Markov Chain within what we call relative coordinates of the latent space. For this architecture, we have convenient access to three subsets of latent variables, $\boldsymbol{\xi}_1, \boldsymbol{\xi}_2$, and $\boldsymbol{\xi}_3$ within absolute coordinates. Fixing $\boldsymbol{\xi}_1$ (resp. fixing $\boldsymbol{\xi}_1$ and $\boldsymbol{\xi}_2$) there is an invertible transformation $h_2(\boldsymbol{\xi}_2; \boldsymbol{\xi}_1)$ (resp. $h_3(\boldsymbol{\xi}_3; \boldsymbol{\xi}_1, \boldsymbol{\xi}_2)$) such that $h_2(\boldsymbol{\xi}_2; \boldsymbol{\xi}_1) \sim \mathcal{N}(0, I)$ (resp. $h_3(\boldsymbol{\xi}_3; \boldsymbol{\xi}_1, \boldsymbol{\xi}_2) \sim \mathcal{N}(0, I)$) under our flow's prior. The Markov Chain in relative coordinates simply follows and proposes transitions for the triplet $(\boldsymbol{\xi}_1, h_2(\boldsymbol{\xi}_2; \boldsymbol{\xi}_1), h_3(\boldsymbol{\xi}_3; \boldsymbol{\xi}_1, \boldsymbol{\xi}_2))$.

As it appears that no test set had been withheld during training of the model, we selected images at random from the full dataset for our experiment. During inference with PL-MCMC, latent space transitions are generated within relative coordinates of the flow's latent space by small perturbations from the current latent state. When perturbing the latent state, proposals are generated following a perturbation kernel, $g_p(\boldsymbol{\xi}'|\boldsymbol{\xi})$, such that $\xi' \sim \mathcal{N}(\boldsymbol{\xi}, \sigma_p^2 I)$. The auxiliary distribution, $q$, is chosen to target $\mathbf{y}_O \sim \mathcal{N}(\mathbf{x}_O, \sigma_a^2 I)$. For this experiment, we select $\sigma_p = 0.01$ and $\sigma_a = 1 \times 10^{-3}$. PL-MCMC is carried out over 25,000 proposals.

### D.3    DETAILS REGARDING CONDITIONAL INFERENCE WITH MNIST DIGITS

In this experiment, we infer missing portions of MNIST (LeCun et al., 1998) digits. The MNIST dataset is composed of $70,000$ monochrome $28 \times 28$ images of handwritten digits. The standard training and test set split for the MNIST dataset is $60,000$ and $10,000$ images, respectively. Our data missingness mechanisms are independent missingness, where pixels are lost uniformly at random, patch missingness, where randomly located contiguous rectangular blocks are missing, and square observation, where only a randomly located contiguous square is observed.

Our chosen normalizing flow is a variant of the NICE (Dinh et al., 2014) architecture. Our implementation is a modification of that by Mu (2019). In the terminology of Kingma & Dhariwal (2018), the model has a depth of flow (K) of 5 and a total of 4 levels (L) and intermediate flow layers have a dimension of 1000. The flow utilizes an independent logistic prior distribution. Rather than splitting even and odd pixels within coupling layers, we split between two randomly selected partitions that are chosen at the time of the flow's initialization and remain fixed for all layers of the flow. Of course, better performance would be expected by selecting a flow architecture that best

suits the spatial organization of image data. However, random partitioning ensures that the expected performance of the flow remains independent of the spatial structuring of the data.

The normalizing flow is trained for $1000$ epochs over the standard $60,000$ element MNIST training set using RMSprop with a learning rate of $1 \times 10^{-5}$ and a momentum of $0.9$ and a batch size of $200$. The data is pre-processed by performing pixel-wise whitening of the dataset (subtracting pixel-wise observed means and dividing by pixel-wise observed standard deviation). The training procedure is intended to follow that used later for training from incomplete MNIST digits to serve as a baseline for comparison and is not intended to produce the best possible generative model of MNIST digits. Examples of unconditioned samples from this model are provided within Figure 13, as obtained with the standard sampling variance (temperature $T = 1.0$, in the terminology of Kingma & Dhariwal (2018))

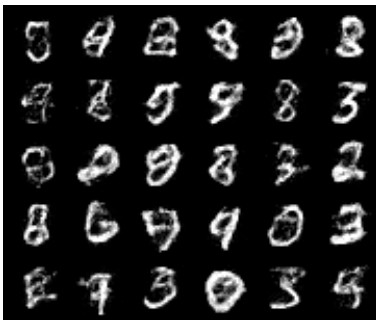

Figure 13: Unconditioned samples at standard variance from MNIST model.

During inference with PL-MCMC, latent space transitions are generated within the absolute coordinates of the flow's latent space, half of the time at random by small perturbations from the current latent state and half of the time entirely resampled at a reduced standard deviation. When perturbing the latent state, proposals are generated following a perturbation kernel, $g_p(\boldsymbol{\xi}'|\boldsymbol{\xi})$, such that $\boldsymbol{\xi}' \sim \mathcal{N}(\boldsymbol{\xi}, \sigma_p^2 I)$. When completely resampling the latent state, proposals are generated following a resampling kernel, $g_r(\boldsymbol{\xi}'|\boldsymbol{\xi})$, such that $\boldsymbol{\xi}' \sim \mathcal{N}(\mathbf{0}, \sigma_r^2 I)$. The auxiliary distribution, $q$, is chosen to target $\mathbf{y}_O \sim \mathcal{N}(\mathbf{x}_O, \sigma_a^2 I)$. For this experiment, we select $\sigma_p = 0.05$, $\sigma_r = 0.5$, and $\sigma_a = 1 \times 10^{-3}$. To simplify calculation of Metropolis-Hastings acceptance probabilities, we employ the assumption that small displacements in the latent space result from $g_p(\boldsymbol{\xi}'|\boldsymbol{\xi})$ while large displacements result from $g_r(\boldsymbol{\xi}'|\boldsymbol{\xi})$, which is valid in the limit of $\sigma_p << \sigma_r$. Therefore, rather than utilize the true transition kernel $g(\boldsymbol{\xi}'|\boldsymbol{\xi})$, we simply assume that $g(\boldsymbol{\xi}'|\boldsymbol{\xi}) \propto g_p(\boldsymbol{\xi}'|\boldsymbol{\xi})$ following a perturbation and $g(\boldsymbol{\xi}'|\boldsymbol{\xi}) \propto g_r(\boldsymbol{\xi}'|\boldsymbol{\xi})$ following a resample. PL-MCMC is carried out over 2,000 proposals. To determine conditional expectations, the results of 20 independent PL-MCMC chains are averaged together.

## E   DETAILS REGARDING QUANTITATIVE EXPERIMENTS

### E.1   DETAILS REGARDING TRAINING FROM INCOMPLETE MNIST DIGITS

In this experiment, we train normalizing flows to model the distribution of MNIST digits from incomplete training data. Our data missingness mechanisms are independent missingness, where pixels are lost uniformly at random, patch missingness, where randomly located contiguous rectangular blocks are missing, and square observation, where only a randomly located contiguous square is observed. For each missingness mechanism, we consider missingness rates of $0.3, 0.6$, and $0.9$. For training, we apply the missingness mechanism to the standard MNIST training set, resulting in a training set of $60,000$ incomplete digits. For testing, we apply the missingness mechanism to the standard MNIST test set, resulting in a test set of $10,000$ incomplete digits.

Our chosen normalizing flow is a variant of the NICE (Dinh et al., 2014) architecture. In the terminology of Kingma & Dhariwal (2018), the model has a depth of flow (K) of 5 and a total of 4 levels (L) and intermediate flow layers have a dimension of 1000. The flow utilizes an independent logistic prior distribution. Rather than splitting even and odd pixels within coupling layers, we split

between two randomly selected partitions that are chosen at the time of the flow's initialization and remain fixed for all layers of the flow. The flow architecture and implementation is the same as used above for training from complete MNIST digits. Better performance could be obtained by selecting an architecture that best suits the spatial organization of image data or by scaling model complexity along with the missingness rate (at a missingness rate of $0.9$, we have a tenth as much observed data available for training as in the complete data case).

In all cases, the normalizing flow is trained for $1000$ epochs using RMSprop with a learning rate of $1 \times 10^{-5}$ and a momentum of $0.9$ and a batch size of $200$. The data is pre-processed by performing pixel-wise whitening of the dataset (subtracting pixel-wise observed means and dividing by pixel-wise observed standard deviation). At each of the first 50 epochs of training, missing pixels are resampled following an independent normal distribution, such that $\mathbf{x}_M \sim \mathcal{N}(\mathbf{0}, I)$ (in whitened coordinates). Every 50 epochs thereafter, missing values are resampled using PL-MCMC as applied to the flow being trained. During inference with PL-MCMC, latent space transitions are generated within the absolute coordinates of the flow's latent space, half of the time at random by small perturbations from the current latent state and half of the time entirely resampled at a reduced standard deviation. When perturbing the latent state, proposals are generated following a perturbation kernel, $g_p(\boldsymbol{\xi}'|\boldsymbol{\xi})$, such that $\boldsymbol{\xi}' \sim \mathcal{N}(\boldsymbol{\xi}, \sigma_p^2 I)$. When completely resampling the latent state, proposals are generated following a resampling kernel, $g_r(\boldsymbol{\xi}'|\boldsymbol{\xi})$, such that $\boldsymbol{\xi}' \sim \mathcal{N}(\mathbf{0}, \sigma_r^2 I)$. The auxiliary distribution, $q$, is chosen to target $\mathbf{y}_O \sim \mathcal{N}(\mathbf{x}_O, \sigma_a^2 I)$. For computation of Metropolis-Hastings probabilities, we employ the same approximation as used when inferring MNIST digits within the qualitative experiments. Throughout training, we use $\sigma_p = 0.05$ and $\sigma_a = 1 \times 10^{-3}$. Within the first 500 epochs, we utilize $\sigma_p = 1.814$. After the first 500 epochs, we resample at reduced variance with $\sigma_p = 0.5$. These parameters and this training procedure were chosen because they provided acceptable performance when applied to training data with the moderate missingness rate of $0.6$. It would certainly be beneficial to determine a more principled approach to their selection. After PL-MCMC sampling, we clamp values between pixel-wise observed minimal and maximal values to produce a newly imputed training set for use in the next 50 epochs of training. Figure 14 below illustrates the progression of how the training set is imputed over training.

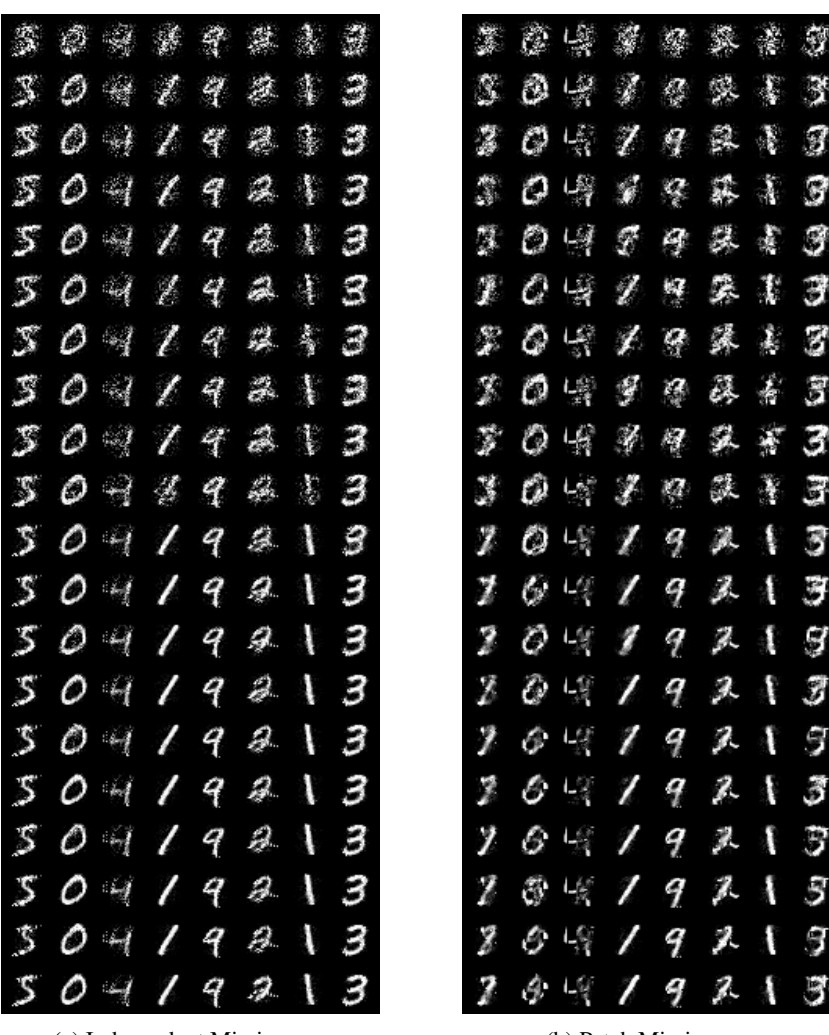

(a) Independent Missingness        (b) Patch Missingness

Figure 14: Example completions of MNIST digit training set by PL-MCMC during MC-EM training at missingness rates are $0.6$. Initial completions are shown in the top row and completions at epoch 950 are shown in the bottom row.

For testing, we utilize the normalizing flows to reconstruct the $10,000$ element incomplete test sets. For this reconstruction, we utilize PL-MCMC chains over $2,000$ proposals following the same procedure as in training. During testing, we use $\sigma_p = 0.05$, $\sigma_p = 0.5$, and $\sigma_a = 1 \times 10^{-3}$. To collect statistics for performance measures, we train three normalizing flows on distinctly prepared (i.e. resampled missingness patterns) training sets, each of which is tested on five distinctly prepared test sets. We consider two methods of employing PL-MCMC to produce reconstructions of missing data. The first method is imputation with a single sample from a PL-MCMC chain (imputation with a sample from the conditional distribution) and the second method is imputation with the average across multiple samples from independent PL-MCMC chains (imputation with the conditional mean). As we average across the samples from 10 independent PL-MCMC chains for the second method, our reported statistics for the first method encompass these additional 10 replications for individual sample imputation performance.

As imputation performance measures, we consider per-pixel reconstruction RMSE and Fréchet Inception Distance (Heusel et al., 2017). To compute Fréchet Inception Distance, we use the implementation provided along with that of MisGAN (Li, 2019). Reconstruction RMSE is recorded within the original, unwhitened representation of pixel data. When imputing $T$ test examples with $\mathcal{M}_i$ denoting the set of pixel indices that are missing for the $i$-th example and $\hat{x}_{i,j}$ denoting our imputed

estimate for the $j$-th pixel of the $i$-th example (having ground truth value $x_{i,j}$), our reconstruction RMSE is calculated as:

$$Reconstruction\ RMSE = \frac{1}{T} \sum_{i=1}^{T} \sqrt{\frac{1}{|\mathcal{M}_i|} \sum_{j \in \mathcal{M}_i} (x_{i,j} - \hat{x}_{i,j})^2}.$$

For comparison, we consider imputation using pixel-wise observed means and imputation using the convolutional variant of MisGAN (Li et al., 2018). We use the implementation of MisGAN provided by Li (2019). The MisGAN models are trained following the provided default parameters (500 epochs with a batch size of 64 with $\tau = 0, \alpha = 0.1, \beta = 0.1, \gamma = 0$, `maskgen = fusion`, `gp_lambda = 10`, `n_critic = 5`, and `n_latent = 128`, with a three layer fully connected imputer network with 784 units in each layer).

### E.2  DETAILS REGARDING TRAINING FROM INCOMPLETE UCI DATASETS

In this experiment, we train normalizing flows to model the distributions of various continuous UCI datasets (Bache & Lichman, 2013) from incomplete training data. In all cases, our data missingness mechanism is independent missingness with a missingness rate of 0.5. A summary of the UCI datasets used in this experiment is provided below in Table 3. For training, we apply the missingness mechanism to the entirety of a single copy of the dataset. For testing, we attempt to reconstruct the missing portions of the training set.

Table 3: Summary of continuous UCI datasets used.

| Dataset | Num. Instances | Num. Attributes |
|---|---|---|
| banknote | 1372 | 4 |
| breast | 569 | 30 |
| concrete | 1030 | 9 |
| red-wine | 1599 | 12 |
| white-wine | 4898 | 12 |
| yeast | 1483 | 8 |

Our chosen normalizing flows are variants of the NICE (Dinh et al., 2014) architecture. In the terminology of Kingma & Dhariwal (2018), all models have a depth of flow (K) of 5 and a total of 4 levels (L) and intermediate flow layers have a dimension of 120. The flows utilize an independent normal prior distribution. Rather than splitting even and odd pixels within coupling layers, we split between two randomly selected partitions that are chosen at the time of the flow's initialization and remain fixed for all layers of the flow. As our implementation works most easily with an even number of attributes, we copy the `concrete` attributes and data missingness to double the number of attributes. By copying data missingness patterns, we ensure that the doubling does not introduce additional information for training. A summary of input attribute dimensions and training batch sizes is provided within Table 4.

Table 4: Summary of continuous UCI datasets used.

| Dataset | Input Dimensions | Batch Size |
|---|---|---|
| banknote | 4 | 3000 |
| breast | 30 | 1500 |
| concrete | 18 | 2000 |
| red-wine | 12 | 3000 |
| white-wine | 12 | 10000 |
| yeast | 8 | 3000 |

In all cases, the normalizing flow is trained for 1000 epochs using Adamax with a learning rate of 0.002, $\beta_1 = 0.9$, and $\beta_2 = 0.999$. We duplicate the training sets (data missingness patterns included) 10 times to form a larger training set without introducing additional information beyond

that present in the original incomplete training set. The batch sizes used are listed in Table 4. The data is pre-processed by performing attribute-wise whitening of the dataset (subtracting attribute-wise observed means and dividing by attribute-wise observed standard deviation). At each of the first 50 epochs of training, missing attribute are resampled following an independent normal distribution, such that $\mathbf{x}_M \sim \mathcal{N}(\mathbf{0}, I)$ (in whitened coordinates). Every 50 epochs thereafter, missing values are resampled using PL-MCMC as applied to the flow being trained. During inference with PL-MCMC, latent space transitions are generated within the absolute coordinates of the flow's latent space, half of the time at random by small perturbations from the current latent state and half of the time entirely resampled at a reduced standard deviation. When perturbing the latent state, proposals are generated following a perturbation kernel, $g_p(\boldsymbol{\xi}'|\boldsymbol{\xi})$, such that $\xi' \sim \mathcal{N}(\boldsymbol{\xi}, \sigma_p^2 I)$. When completely resampling the latent state, proposals are generated following a resampling kernel, $g_r(\boldsymbol{\xi}'|\boldsymbol{\xi})$, such that $\boldsymbol{\xi}' \sim \mathcal{N}(\mathbf{0}, \sigma_r^2 I)$. The auxiliary distribution, $q$, is chosen to target $\mathbf{y}_O \sim \mathcal{N}(\mathbf{x}_O, \sigma_a^2 I)$. For computation of Metropolis-Hastings probabilities, we employ the same approximation as used when inferring MNIST digits within the qualitative experiments. Throughout training, we use $\sigma_p = 0.01$, $\sigma_r = 1.0$, $\sigma_a = 1 \times 10^{-3}$. These parameters and this training procedure were chosen because they provided acceptable performance across the UCI datasets considered. It would certainly be beneficial to determine a more principled approach to their selection. After PL-MCMC sampling, we clamp values between attribute-wise observed minimal and maximal values to produce a newly imputed training set for use in the next 50 epochs of training.

For testing, we utilize the normalizing flows to reconstruct the missing values from their training sets. For this reconstruction, we utilize PL-MCMC chains over $2,000$ proposals following the same procedure as in training. During testing, we use $\sigma_p = 0.01$, $\sigma_p = 1.0$, and $\sigma_a = 1 \times 10^{-3}$. To collect statistics for performance measures, we train five normalizing flows on distinctly prepared (i.e. resampled missingness patterns) training sets. We consider two methods of employing PL-MCMC to produce reconstructions of missing data. The first method is imputation with a single sample from a PL-MCMC chain (imputation with a sample from the conditional distribution) and the second method is imputation with the average across multiple samples from independent PL-MCMC chains (imputation with the conditional mean). As we average across the samples from 25 independent PL-MCMC chains for the second method, our reported statistics for the first method encompass these additional 25 replications for individual sample imputation performance. In the case of the `concrete` dataset, we consider the copied attribute values as an additional single sample from the conditional distribution that is also incorporated into averaging.

As an imputation performance measure, we consider per-attribute normalized MSE. When imputing $T$ test examples with $\mathcal{M}_i$ denoting the set of attribute indices that are missing for the $i$-th example and $\hat{x}_{i,j}$ denoting our imputed estimate for the $j$-th attribute of the $i$-th example (having ground truth value $x_{i,j}$), our normalized MSE is calculated as:

$$NMSE = \frac{1}{T} \sum_{i=1}^{T} \frac{1}{|\mathcal{M}_i|} \sum_{j \in \mathcal{M}_i} (\frac{x_{i,j} - \hat{x}_{i,j}}{\sigma_j})^2,$$

where $\sigma_j$ denotes the ground truth standard deviation of the $j$-th attribute.

For comparison, we consider imputation using attribute-wise observed means, imputation using the missForest (Stekhoven & Bühlmann, 2012) R package with default parameters, and imputation with VAEs using MIWAE (Mattei & Frellsen, 2019). For imputation with missForrest, no data preprocessing is employed. We use the implementation of MIWAE provided by Mattei (2019). In all cases, the VAE architecture employed has an intrinsic dimension $d$ of 10, an encoder and decoder comprised of 3 layers each with 128 hidden units with ReLU activation functions, an independent normal prior, and a Student's $t$ distribution observation model. In all cases, we utilize zero imputation as the MIWAE imputation function. For training, we use 20 importance weights while for inference we use $10,000$ importance weights. We train the models using the provided default parameters ($2,000$ epochs using Adam with a learning rate of $0.001$ and a batch size of $64$). In all cases, the data is pre-processed by performing attribute-wise whitening of the dataset (subtracting attribute-wise observed means and dividing by attribute-wise observed standard deviation).

### E.3 DETAILS REGARDING SAMPLING EFFICIENCY FOR INFERENCE OF MNIST DIGIT

In these experiments, we use MCMC to estimate the conditional expectation for the missing portion of a single MNIST digit. In this case, the data missingness mechanism is a checkerboard pattern masking half of the digit, as shown in Figure 15.

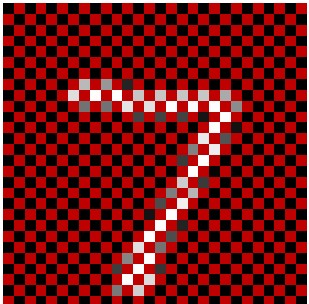

Figure 15: Masked digit used for sampling efficiency experiments.

To perform this inference, we use the same normalizing flow as used in the qualitative experiments in Section 5.3, detailed within Appendix D.3. During inference with PL-MCMC, latent space transitions are generated within the absolute coordinates of the flow's latent space, half of the time at random by small perturbations from the current latent state and half of the time entirely resampled at a reduced standard deviation. When perturbing the latent state, proposals are generated following a perturbation kernel, $g_p(\boldsymbol{\xi}'|\boldsymbol{\xi})$, such that $\xi' \sim \mathcal{N}(\boldsymbol{\xi}, \sigma_p^2 I)$. When completely resampling the latent state, proposals are generated following a resampling kernel, $g_r(\boldsymbol{\xi}'|\boldsymbol{\xi})$, such that $\boldsymbol{\xi}' \sim \mathcal{N}(\mathbf{0}, \sigma_r^2 I)$. The auxiliary distribution, $q$, is chosen to target $\mathbf{y}_O \sim \mathcal{N}(\mathbf{x}_O, \sigma_a^2 I)$. For this experiment, unless otherwise specified, we select $\sigma_p = 0.01$, $\sigma_r = 1.0$, and $\sigma_a = 1 \times 10^{-3}$. For computation of Metropolis-Hastings probabilities, we employ the same approximation as used when inferring MNIST digits within the qualitative experiments.

Ideally, for comparison with PL-MCMC, we would consider employing a Metropolis-Hastings MCMC through the modeled data space proposing $\mathbf{x}_M' \sim \mathcal{N}(\mathbf{x}_M, \sigma_{MH}^2 I)$, for some appropriately chosen $\sigma_{MH}$. However, we found that naive Metropolis-Hastings through the modeled data space required such a small $\sigma_{MH}$ that the Markov Chain made no discernible progress. We therefore resorted to per-missing-pixel Gibbs sampling, with missing pixel proposals generated following $x_{M,i}' \sim \mathcal{N}(\mu_i, \sigma_i^2)$, with $\mu_i$ and $\sigma_i$ denoting the $i$-th missing pixel's observed mean and standard deviation throughout the training set.

For both techniques, the initial state of Markov Chain (latent state for PL-MCMC and missing pixel values for Gibbs sampling) is determined by unconditional sampling from the model at standard variance (temperature $T = 1.0$, in the terminology of Kingma & Dhariwal (2018)). In each replication, the conditional mean is estimated from the average of 100 independent Markov Chains. To gather statistics regarding the variance of estimated conditional mean RMSE, we perform 10 separate replications of the experiment.

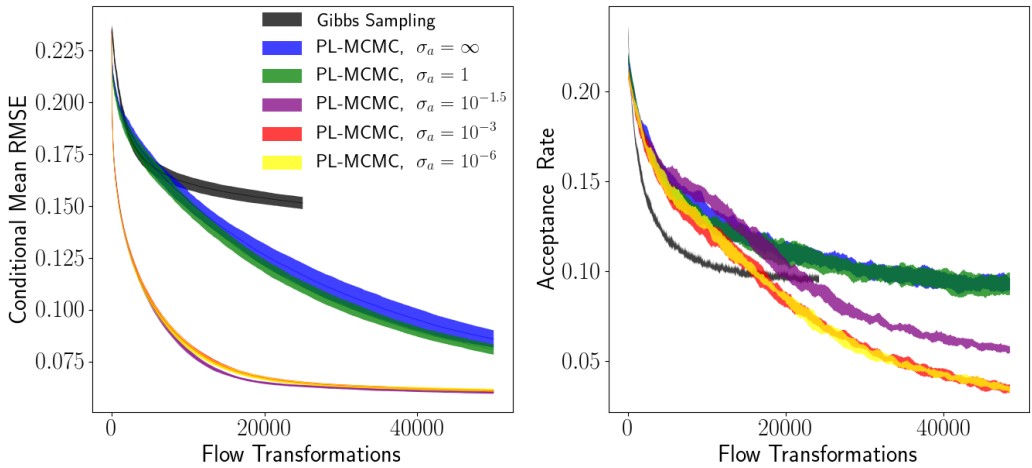

Figure 16: Single standard deviation envelopes of estimated conditional mean per-pixel RMSE and proposal acceptance rate for conditional sampling of MNIST digit. PL-MCMC implementations only differ by choice of auxiliary density.

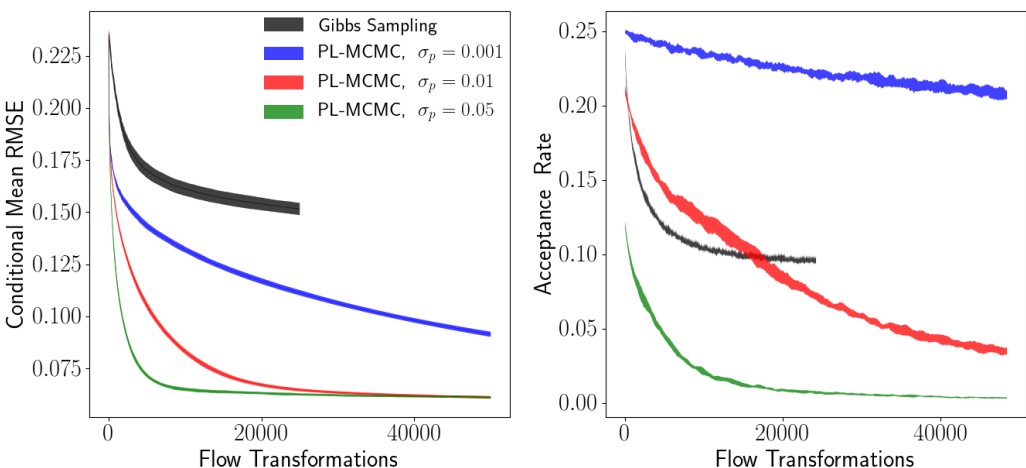

Figure 17: Single standard deviation envelopes of estimated conditional mean per-pixel RMSE and proposal acceptance rate for conditional sampling of MNIST digit. PL-MCMC implementations only differ by the scale of perturbations used in their transition proposals.

As the evaluation of a PL-MCMC Metropolis-Hastings proposal involved two transformations through the normalizing flow while the evaluation of a Gibbs sampling proposal involves only one transformation, it can be argued that each PL-MCMC proposal was twice as costly as each Gibbs sample. For this reason, Figures 16 and 17 perform the same comparisons as Figures 7 and 8, but with respect to the number of flow transformations utilized in the Markov Chains. We can see that, even accounting for the relative computational costs of the two methods, PL-MCMC offers significantly improved sampling performance.

Figures 18 and 19 compare the effect of altering proposal distribution scale when the transition proposals are generated only by the perturbation kernel. For reference, the results from our default $\sigma_p = 0.01$, $\sigma_r = 1.0$, and $\sigma_a = 1 \times 10^{-3}$ PL-MCMC implementation are also included in red.

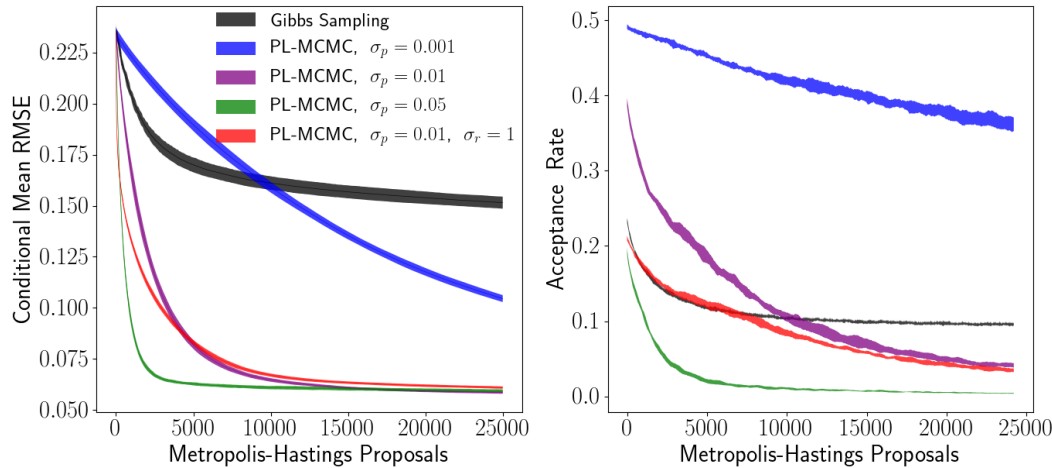

Figure 18: Single standard deviation envelopes of estimated conditional mean per-pixel RMSE and proposal acceptance rate for conditional sampling of MNIST digit. Unless otherwise stated, PL-MCMC implementations only utilize a perturbation transition kernel and only differ by the scale of perturbations used in their transition proposals

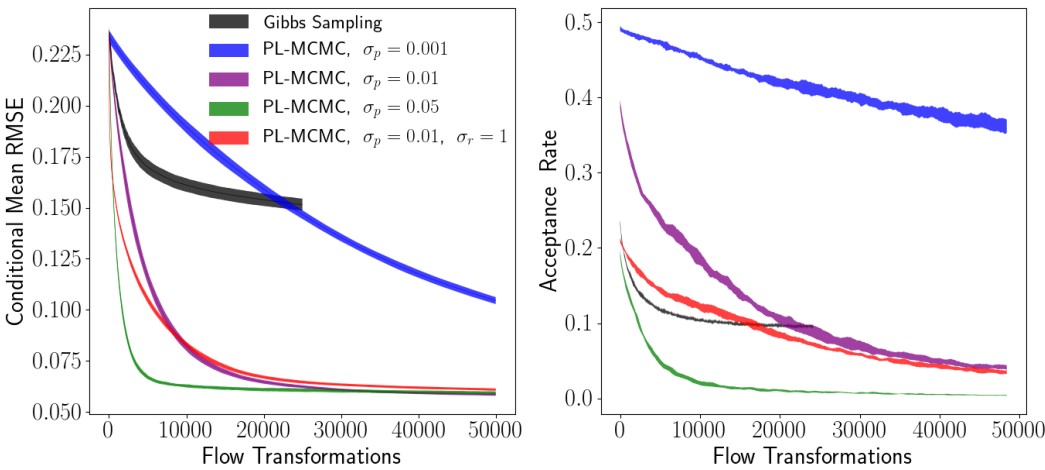

Figure 19: Single standard deviation envelopes of estimated conditional mean per-pixel RMSE and proposal acceptance rate for conditional sampling of MNIST digit. Unless otherwise stated, PL-MCMC implementations only utilize a perturbation transition kernel and only differ by the scale of perturbations used in their transition proposals.

Figures 20 and 21 compare the effect of altering the scale of the transition proposal's resampling kernel.

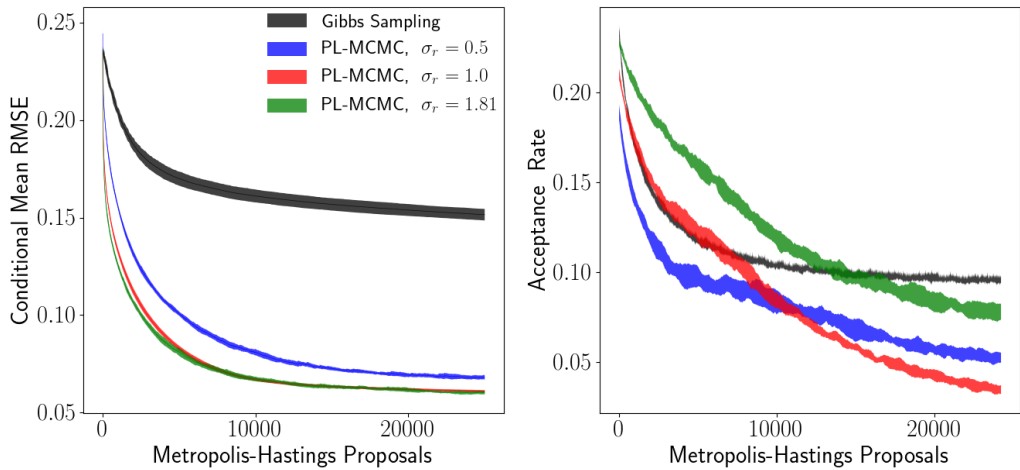

Figure 20: Single standard deviation envelopes of estimated conditional mean per-pixel RMSE and proposal acceptance rate for conditional sampling of MNIST digit. PL-MCMC implementations only differ by the scale of the resampling kernel for their transition proposals.

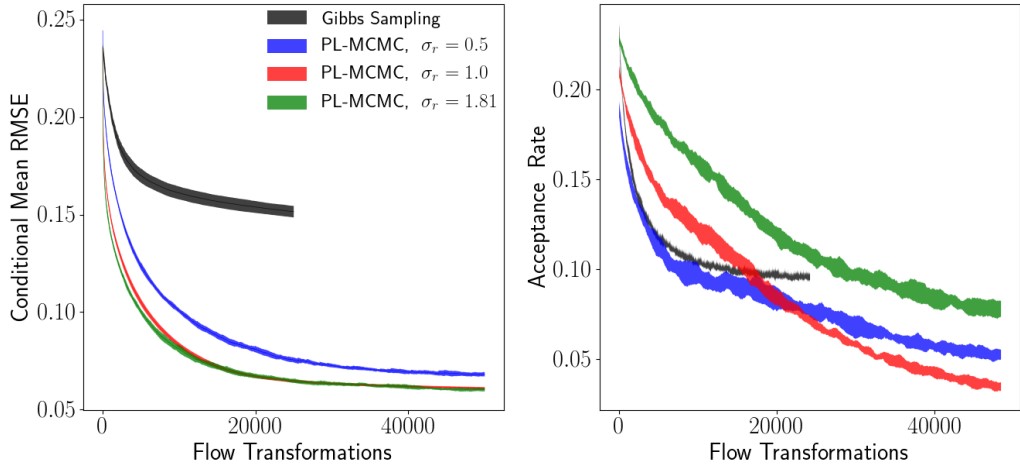

Figure 21: Single standard deviation envelopes of estimated conditional mean per-pixel RMSE and proposal acceptance rate for conditional sampling of MNIST digit. PL-MCMC implementations only differ by the scale of the resampling kernel for their transition proposals.

### E.4 RESTRICTIVE AUXILIARY DISTRIBUTIONS DO NOT NECESSARILY REDUCE PL-MCMC TO STOCHASTIC SEARCH

Given that our results from auxiliary distributions with standard deviations of $\sigma_a = 10^{-3}$ and $\sigma_a = 10^{-6}$ closely overlap, we may be concerned that the auxiliary distribution might dominate PL-MCMC's behavior and reduce the procedure to a simple search in the latent space to best rebuild the observed data. To determine whether an auxiliary distribution with $\sigma_a = 10^{-3}$ overwhelmingly dominates the conditional sampling process, we follow a PL-MCMC implementation with $\sigma_a = 10^{-3}$ and determine how often the its decisions regarding proposal acceptance would be contradicted by an implementation with $\sigma_a = \infty$. These results are provided within Figure 22.

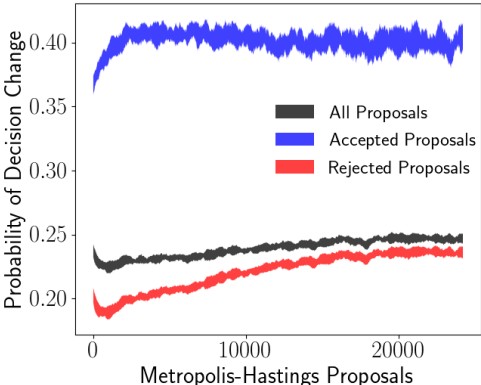

Figure 22: Probabilities that a PL-MCMC implementation with $\sigma_a = 10^{-3}$ makes proposal decisions that would be contradicted by an implementation with $\sigma_a = \infty$. "Accepted" and "Rejected" proposals refer to the decisions made by the $\sigma_a = 10^{-3}$ implementation.

Fundamentally, we see that, at least with $\sigma_a = 10^{-3}$ in this particular task, the two implementations are likely to agree in their decisions to accept or reject particular proposal transitions. Choosing an auxiliary distribution with $\sigma_a = 10^{-3}$ does not overwhelmingly "bully" the Markov Chain into mindlessly reconstructing observed data. Still, there is a substantial probability that this choice of auxiliary distribution will alter the decision, which is the mechanism by which the auxiliary distribution helps to guide the Markov Chain and improve conditional sampling performance. We suspect that these decision change probability computations could be useful for tuning the choice of auxiliary distribution.

