# OpenReview forum: "Projected Latent Markov Chain Monte Carlo: Conditional Sampling of Normalizing Flows"
_ICLR.cc/2021/Conference — ICLR 2021 Poster_

### Official Review · AnonReviewer4 · 2020-10-26

**Rating:** 6
**Confidence:** 4

**Review:**

Summary:
The submission introduces an approach for conditional sampling by running a Random walk Metropolis algorithm on the latent space as a pullback of normalizing flows. The normalizing flows are learnt via a MC-EM approach for incomplete training data. The method is illustrated on numerous experiments sampling missing parts for image and UCI datasets.

Positives:
The combination of normalizing flows and MCMC sampling for conditional sampling is new as far as I am aware and is an interesting approach. Qualitative experimental results on different data sets seem promising with quantitative results indicating that it performs competitively (better reconstruction RMSE compared to MisGAN, closely matching the performance of MIWAE). More advanced MCMC samplers (such as gradient-based ones) could be used in the proposed framework, potentially yielding better experimental results.

Negatives:
The approach relies on hyperparameters such as the choice of the auxiliary data density q and the MCMC proposal density g, and I feel that choices for these should be analyzed in greater detail in the paper. For example, the choice of both q and g affect the exploration of the target and so do they affect the sampling quality in practice for a reasonable number MCMC steps? Likewise, it is well documented that adaptive MCMC methods can converge to a different target and I am not yet convinced that the MC-EM approach approximately maximizes the complete training data likelihood without some restrictions on the MCMC adaptation strategy.

Recommendation:
I am recommending a weak accept, since the approach is interesting and is supported by many experiments, however, the hyperparameter choices for the adaptation should be better analyzed.

Comments:
How do you choose the perturbation scales for the random walk proposals?
Could it be that the target has different scales across different dimensions, so that the MCMC kernel explores only some dimensions well and does this affect the learning process?
Do the acceptance rates have a large influence on the MC-EM training? My intuition would be that for high acceptance rates, the learning process would be more volatile (the training samples X’_train would have many same entries and some very different ones)?
Why do the acceptance rate go to zero in Figure 7?
Why do you make the approximation of the Metropolis-Hastings ratio in the experiments? Is the proposal not just a simple 2-mixture density which should not incur a much higher computational cost?

---

> ### Author Response · Authors · 2020-11-15
> **Excellent suggestion regarding hyperparameter analysis.  One potential clarification.**
>
> Thank you for your time and consideration.
>
> To address a potential misunderstanding, our proposed technique only requires the normalizing flow to satisfy very mild positivity and smoothness constraints to ensure asymptotically exact conditional sampling.  The described MC-EM training procedure is not required for the conditional sampling from normalizing flows.  We included the MC-EM training procedure as a notable example application of the PL-MCMC technique.  The experiments involving the MC-EM training of normalizing flows are intended to offer some indication as to how effectively PL-MCMC samples from its intended distributions.  These experiments are not intended to advocate for our particular implementation of the MC-EM training procedure.
>
> To address individual comments:
>
> * **How do the choices of auxiliary distribution, $q$, and transition proposal density, $g$, affect conditional sampling performance?**
>   In the revised draft of our work, we have expanded Section 5.3 and Appendix E.3 to better explore the effect choices of $q$ and $g$ have on conditional sampling performance with PL-MCMC.
>
> * **Does the implemented MC-EM training procedure maximize complete data likelihood?**
>
>   We assume the concern is whether the MC-EM training procedure maximizes complete data likelihood, in the sense of maximizing the likelihood of the complete ground truth data if training on an intentionally corrupted data set.  In this case, we point out that we do not believe the MC-EM training procedure should be used to train a normalizing flow from a complete data training set by intentionally removing values from the training data.  If a complete data training set is available, the normalizing flow should be trained to maximize the complete data likelihood directly.  For conditional sampling, PL-MCMC does not require the flow to have been trained in any particular manner to ensure asymptotically exact sampling from the flow's described distributions.
>
>     If this concern were instead whether our implemented MC-EM procedure maximizes observed data likelihood (the intended objective of expectation-maximization), we would agree that this may not be guaranteed.  We would point out that we made no claim of such a guarantee and that we do not intend to advocate for our particular implementation of the MC-EM procedure.  We simply use the comparative success of our MC-EM training to determine whether PL-MCMC can accurately sample from its intended conditional distributions.
>
> * **How do you choose the perturbation scales for transition proposals?**
>
>   Because this work is intended to introduce and prove a novel technique, we only sought to demonstrate that there exist some choices of these transition proposals that enable efficient conditional sampling.  The scales presented within the work were therefore determined by some manual trial and error across some reasonable values.  We leave the development of a principled method for selecting these transition proposal scales to future work.
>
> * **Can the natural scale of the target distribution differ across dimensions and affect approval rate?**
>
>   The fixed proposal density Metropolis-Hastings implementation PL-MCMC can absolutely be affected by the target distribution scaling unevenly across different dimensions.  This issue arises from the Metropolis-Hastings implementation, and not from PL-MCMC itself.  We leave the development and proof of more advanced and adaptive implementations of PL-MCMC to future work.
>
> * **How does approval rate influence MC-EM training volatility?**
>
>   In our view, the approval rate reflects how well matched the proposal density is to the target distribution.  Markov Chains can have very low mixing rates (hence low volatility) while maintaining high acceptance rates.  We believe that volatility during MC-EM training (assuming the conditional sampling remains reasonably satisfactory) is derived from the properties of the normalizing flow's conditional distributions.  If the flow's conditional distribution admits many likely completions for missing values, then the training could be more volatile.
>
> * **Why does the acceptance rate fall in Figure 7?**
>
>  This drop is the result of our using a constant scale transition distribution.  Performance in this regard could be improved by the future development of adaptive implementations of PL-MCMC.
>
> * **Why do you use an approximation for transition probability when including a resampling kernel?**
>
>   With our first implementation utilizing the exact transition densities, we ran into numerical issues when calculating the ratio of transition densities.  The approximation nicely avoided these numerical issues while introducing a negligible error for the transition distributions considered within our experiments.

---

### Official Review · AnonReviewer1 · 2020-10-28
**Good paper on missing data completion with normalizing flows, some questions**

**Rating:** 7
**Confidence:** 4

**Review:**

I enjoyed reading this paper and the idea of combining normalizing flow density models with conditional sampling seems natural, useful and has interesting potential applications to missing data problems.

Things that were not clear to me are the following:

1. I don't understand what this "pre-trained" model being referred to is. Is this a generative model for the entire data set? Do you mean that you get the model $p_{f, \theta}$ from somewhere already and not learn it yourself?

2. The choice of $q$ is one of the more interesting parts of the proposed algorithm that deserves to be studied in greater detail! If we don't use $q$, wouldn't the algorithm still work? You should try a version of your algorithm without assuming any $q$ on $y_O$, then checking how much does it affect performance.

3. I am not convinced that it is fair to set "q to be an independent normal distribution centered on the conditioning values $x_{O}$". Clearly, with sufficiently low variance for $q$ it is possible to force the chain to sample values of $y_{O}$ that are very close to $x_{O}$. Since the dependence between $y_{O}$ and $y_{M}$ is generally strong, this will have you perform good reconstruction of $x_{M}$. If you do use a $q$ you should try variances other than the (1e-03)^2 which you have used as mentioned in the Appendix.

4. Is it expensive to compute the probability density for the Metropolis correction? If you don't do a Metropolis correction, do the conditional samples still look reasonably good?

5. Somewhat related to 2). In section 6.3 figure 7 the acceptance rate for PL-MCMC looks really low and is continuing to drop. You should probably run the chain for even longer until the acceptance rate plateaus completely and see what the samples at this point look like. Or perhaps you should decrease your proposal scaling.

I am willing to upgrade my review if particularly point 2,3 is addressed.

---

> ### Author Response · Authors · 2020-11-15
> **Great questions and suggestions.**
>
> Thank you for your time and consideration.
>
> For each question in order:
>
> 1. **What is meant by "pre-trained" model?**
>   By "pre-trained" model, we mean a publicly available model that we have not trained ourselves.  One of the motivations for PL-MCMC is to enable efficient conditional sampling from all of the impressive normalizing flows that have recently been introduced.  It is not unreasonable to suppose that certain architectures or training methodologies could yield a normalizing flow that is better suited to conditional sampling using PL-MCMC.  We hope that our use of these "pre-trained" models provides a relatively faithful indication of the performance that can be expected of PL-MCMC to perform conditional inference with the normalizing flows that are already trained and in use today.
>
> 2. **Can the auxiliary density, $q$, be omitted?**
>
>   We believe not using $q$ is equivalent to replacing $q$ with an improper uniform distribution.  Our proof of the implementation's convergence in Section 3.3 requires $q$ to be a proper and positive density.  Because convergence is guaranteed for all proper choices of $q$ in a limiting sequence towards the improper uniform distribution (e.g. $q = \mathcal{N}(\mathbf{x}_{O}; \sigma I)$ as $\sigma \rightarrow \infty$), we suspect that $q$ can be omitted without losing asymptotic convergence.  We have included this choice of $q$ in our revised expansion of Section 5.3.  We find that omitting $q$ still leads to improved sampling performance,due to PL-MCMC's use of the flow's latent space to produce effective transition proposals, but find that "stronger" choices of $q$ can provide even better sampling performance.
> 3. **Is it fair to make the auxiliary distribution, $q$, strongly centered on the observed data?**
>
>   Selecting $q$ to be a multivariate normal distribution centered on the observed data values offers no additional information beyond what is already available within the context of conditional sampling given the observed values.  If analytic conditional sampling from normalizing flows were available to us, we would not consider it's use of the observed data values unfair and we simply call the process "conditional sampling from the model described by the normalizing flow".  Without a tractable analytical method, we are forced to use a Markov Chain technique, wherein the auxiliary distribution, $q$, encourages the Markov Chain to favor exploring latent states that rebuild the observed data.
>
>     Our current intuition regarding $q$ is that it represents some prior belief regarding the coupling between observed and missing data within the model described by the normalizing flow.  If, as is often the case, this prior belief is correct and the normalizing flow does exhibit tight coupling between missing and observed data, then choosing $q$ to be strongly centered on observed data will benefit conditional sampling performance with PL-MCMC.  If it happens that the prior belief associated with "strong" choice of $q$ is incorrect, the results are not disastrous, as asymptotic convergence is still guaranteed, but sampling performance could be worse than would be obtained with a "softer" choice of $q$.  When PL-MCMC exhibits good performance in the task of reconstructing missing data, its performance derives entirely from the computational effort originally expended in training an accurate normalizing flow with a convenient-to-explore latent space.
>
> 4. **What is the computational cost of PL-MCMC?**
>
>   For normalizing flows, the cost of evaluating joint likelihood is dominated by the cost of performing the transformation from the flow's modeled data space to its latent space.  For the popular normalizing flow architectures that we are aware of, this transformation is as costly as its reverse transformation.  PL-MCMC requires one transformation from  the latent space to the modeled data space and one transformation from the modeled data space to the latent space.  We estimate that a Metropolis-Hastings implementation of PL-MCMC is about twice as costly as any Metropolis-Hastings technique operating within the modeled data space of the flow.  The potential sampling performance increase provided by PL-MCMC can far outweigh this doubling of computational cost per Metropolis-Hastings proposal.
>
>   We would strongly advise against dropping the correction term, as this removes the guarantee of asymptotic convergence while only giving a modest doubling of speed.
>
> 5. **On the fall of acceptance rate within Figure 7:**
>
>   This drop is  the result of our using a constant scale transition distribution.  Performance in this regard could be improved by the future development of adaptive implementations of PL-MCMC.  We investigate the effect of reducing proposal scale within our expanded analysis regarding the effect of $q$ and $g$ on conditional sampling performance in Section 5.3.

---

### Official Review · AnonReviewer3 · 2020-10-29
**A slight variation on standard stochastic EM algorithms**

**Rating:** 6
**Confidence:** 3

**Review:**

Summary

This paper proposes a stochastic expectation--maximisation (EM) algorithm. The main idea is that the target distribution is specified as a deterministic mapping, a.k.a. a normalising flow, from some simple "base" distribution.


Strengths

The algorithm appears to be formally correct (in the sense that it is a standard stochastic EM algorithm). The method is demonstrated on a large number of examples.

Weaknesses

The proposed algorithm is just a standard Metropolis--Hastings (MH) update interspersed with a stochastic EM update for the parameters of the target distribution. This does not seem novel; such algorithms have been around for decades.

The authors should explain why they need for extending the space to include $y_O$ instead of just mapping $\xi$ to $y_M$.


Minor comments

- There are a number of typos in the bibliography mostly related to inconsistent use of capital letters in article titles and journal/conference names.

- What does the semi-colon in $p_{f,\theta}(x_M; x_O)$ mean? Why not use a comma if this is meant to be a joint density?

- In Section 3, it would be helpful to write the (extended) target distribution of the Metropolis--Hastings algorithm down explicitly and formally.

- Within LaTeX's maths mode, you cannot just write operators $min$ and $Uniform$ (LaTeX treats this, e.g., as multiplying $m$ by $i$ by $n$ which leads to the wrong spacing).

---

> ### Author Response · Authors · 2020-11-15
> **The contribution of the work is on conditional sampling, not EM training.**
>
> Thank you for your time and consideration.
>
> To address a misunderstanding, the purpose of our work (as indicated throughout) is to introduce a method for efficiently sampling from the conditional distributions known by normalizing flows.  The described EM training procedure is not the focus of the work.  Instead, the EM training procedure is described (see the beginning of Section 4) as an application of our proposed conditional sampling methodology.  We explain (see the first paragraph of Section 6) that the purpose of our experiments regarding the EM training of normalizing flows from missing data is to gauge whether our proposed conditional sampling methodology effectively samples from its intended distributions in practice.  In our discussion of the results of our EM training experiments (see the final paragraphs of Sections 5.1 and 5.2), we conclude that these experiments demonstrate that our proposed methodology is at least sufficiently effective in sampling from the conditional distributions of normalizing flows to enable the EM training of normalizing flows from missing data to a standard comparable to that established by existing techniques for training other generative models from incomplete data.
>
> To address particular comments:
>  * **Why extend the Markov Chain to include the working variable $\\mathbf{y}_{O}$?**
> This extension of the Markov Chain's state space is fundamental to the operation of our proposed PL-MCMC conditional sampling technique.  Including $\\mathbf{y}_{O}$ allows us to define a Markov Chain within the latent space of the normalizing flow that is asymptotically guaranteed (see Section 3.3) to sample from the desired conditional distributions.  If the state space is not expanded, asymptotically exact conditional sampling requires solving intractable integrals relating to conditional likelihoods given the observed data.  Expanding the state space lets us leverage the latent representation already provided by the normalizing flow while avoiding otherwise intractable computations.
> * **What is the meaning of the semi-colon within $p_{f, \\theta}$ $(\\mathbf\{x\}_\{M\}; \\mathbf\{x\}_\{O\})$?**
>
>  By $p_{f, \\theta}$ $(\\mathbf\{x\}_\{M\}; \\mathbf\{x\}_\{O\})$, we are referring to the normalizing flow's modeled joint density between the missing and observed portions of the modeled data.  We find that using the traditional comma notation for the joint density heavily implies a fixed ordering for the density function's arguments that conflicts with the subsets of missing and observed variables.  We use the semi-colon notation to represent the somewhat unusual operation of separating observed from missing variables.  While the notation may not be particularly common, we believe it remains unobtrusive and concisely represents the underlying mathematical concept.
> * **Description of PL-MCMC's target distribution:**
>
>  A complete description of the target distribution was already provided within Section 3.1, "The Projected Latent Target Distribution".
> * **Comments regarding typesetting:**
>
>  We have addressed these typographic problems within the newly revised version of the work.

---

### Decision · Program_Chairs · 2021-01-07
**Final Decision**

**Decision:**

Accept (Poster)

**Comment:**

This work combines normalizing flows with conditional sampling. While there are connections to other works, the paper seems novel and applicable, and has nice experimental results. The authors did a good job clarifying the reviewers questions, and have addressed their major concerns. We appreciate the additional analyses added to the paper.